# Opponent learning with different representations in the cortico-basal ganglia pathways can develop obsession-compulsion cycle

Reo Sato[1]☯, Kanji Shimomura[1]☯, Kenji Morita[1,2]*

**1** Physical and Health Education, Graduate School of Education, The University of Tokyo, Tokyo, Japan,
**2** International Research Center for Neurointelligence (WPI-IRCN), The University of Tokyo, Tokyo, Japan

☯ These authors contributed equally to this work.
* morita@p.u-tokyo.ac.jp

**Data Availability Statement:** All relevant codes are within the Supporting Information files. Codes to generate/reproduce the data presented in the

## Abstract

Obsessive-compulsive disorder (OCD) has been suggested to be associated with impairment of model-based behavioral control. Meanwhile, recent work suggested shorter memory trace for negative than positive prediction errors (PEs) in OCD. We explored relations between these two suggestions through computational modeling. Based on the properties of cortico-basal ganglia pathways, we modeled human as an agent having a combination of successor representation (SR)-based system that enables model-based-like control and individual representation (IR)-based system that only hosts model-free control, with the two systems potentially learning from positive and negative PEs in different rates. We simulated the agent's behavior in the environmental model used in the recent work that describes potential development of obsession-compulsion cycle. We found that the dual-system agent could develop enhanced obsession-compulsion cycle, similarly to the agent having memory trace imbalance in the recent work, if the SR- and IR-based systems learned mainly from positive and negative PEs, respectively. We then simulated the behavior of such an opponent SR+IR agent in the two-stage decision task, in comparison with the agent having only SR-based control. Fitting of the agents' behavior by the model weighing model-based and model-free control developed in the original two-stage task study resulted in smaller weights of model-based control for the opponent SR+IR agent than for the SR-only agent. These results reconcile the previous suggestions about OCD, i.e., impaired model-based control and memory trace imbalance, raising a novel possibility that opponent learning in model(SR)-based and model-free controllers underlies obsession-compulsion. Our model cannot explain the behavior of OCD patients in punishment, rather than reward, contexts, but it could be resolved if opponent SR+IR learning operates also in the recently revealed non-canonical cortico-basal ganglia-dopamine circuit for threat/aversiveness, rather than reward, reinforcement learning, and the aversive SR + appetitive IR agent could actually develop obsession-compulsion if the environment is modeled differently.

figures are available at:https://github.com/
kenjimoritagithub/sr101

**Funding:** KM was supported by Grant-in-Aid for
Scientific Research (No. 20H05049 and
23H03295) of the Ministry of Education, Culture,
Sports, Science and Technology in Japan (MEXT)
(http://www.mext.go.jp/en/) and the Japan Society
for the Promotion of Science (JSPS) (https://www.
jsps.go.jp/english/) and the Naito Foundation
(https://www.naito-f.or.jp/en/). The funders had no
role in study design, data collection and analysis,
decision to publish, or preparation of the
manuscript.

**Competing interests:** The authors have declared
that no competing interests exist.

## Author summary

Obsessive-compulsive disorder (OCD) is one of the major psychiatric disorders diagnosed
in 2.5%-3% of the population, and is characterized as an enhanced cycle of obsessive
thought, e.g., whether the door was locked, and compulsive action, e.g., checking door
lock. It remains elusive why such an apparently maladaptive behavior could be enhanced.
A prevailing theory proposes that humans use two control systems, flexible yet costly
goal-directed system and inflexible yet costless habitual system, and impairment of the
goal-directed system leads to OCD. On the other hand, recent work proposed a new the-
ory that shorter memory trace for credit-assignment of negative, than positive, prediction
errors can induce OCD. Relation between these two theories remains unclear. We show
that opponent learning of particular type of goal-directed(-like) system, suggested to be
implemented in the brain, and habitual system from positive versus negative prediction
errors could exhibit an (apparent) overall decrease in goal-directed control and also
develop enhanced obsession-compulsion cycle similar to the one developed by memory-
trace imbalance, thereby bridging the two theories. Such an opponent learning of the two
systems was actually suggested to be advantageous in certain dynamic environments, and
could thus be evolutionarily selected at the cost of possible development of OCD.

## Introduction

Obsessive-compulsive disorder (OCD) is one of the major psychiatric disorders diagnosed in
2.5%-3% of the population [1], and is characterized as an enhanced cycle of obsessive thought,
e.g., whether the door was locked, and compulsive action, e.g., checking door lock. It remains
elusive why such an apparently maladaptive behavior could be enhanced. Dual process theo-
ries and their advanced forms [2–4] suggest that humans use two control systems, one flexible
yet costly goal-directed/model-based system, where the "model" refers to the internal model of
the environments (state transitions and rewards/punishments), and another inflexible yet cost-
less habitual/model-free system. A prevailing suggestion based on these theories is that
impairment of the goal-directed/model-based system relates to OCD [1,5–7], potentially
explaining its apparently irrational behavior. In the two-stage decision task with reward or no-
reward outcomes [8], OCD patients showed impaired model-based choices [6,7]. Also, deficits
in model-based control in the two-stage task were strongly associated with a symptom dimen-
sion "compulsive behavior and intrusive thought" in the general population [9].

Meanwhile, recent work [10] suggested that memory trace imbalance underlies OCD. Spe-
cifically, the authors constructed an environmental model that describes potential enhance-
ment of obsession-compulsion cycle, and showed that such a cycle can be enhanced in the
agent having much shorter memory (eligibility) trace for negative than positive reward predic-
tion errors (RPEs). Then, they conducted fitting of the behavioral choice in a delayed feedback
task, showing that OCD patients were indeed fitted with such imbalanced eligibility traces as
compared to healthy controls (HCs), although even some HCs tended to show a similar imbal-
ance. Mechanisms of such memory trace imbalance, as well as its relation to the suggestion of
impaired model-based control, remain unclear.

Whereas the abovementioned modeling in the recent work [10] examined the agent that
only had model-free control, multiple studies have suggested that humans and other animals
use both model-based and model-free control [4,8]. Model-free control has been suggested to
be implemented in the way that dopamine represents RPEs and dopamine-dependent

plasticity of cortico-striatal synapses represents RPE-based update of state/action values [11,12]. Neural implementation of model-based control has remained more elusive, but recent studies suggest that if states or actions are represented in a manner that contains information about action-state transitions, partially model-based-like control can be acquired through RPE-based update and thereby through dopamine-dependent plasticity, similarly to model-free control [13]. As such a representation, successor representation (SR), in which state/action is represented by cumulative occupancies of successor states/actions (Fig 1A) [14], has been suggested to be used in humans [15] and implemented in the brain [16–18].

Different cortical regions or neural populations may adopt different ways of representations, in particular, SR and more classical individual (punctate) representation (IR), the latter of which can implement model-free control. These different cortical regions/populations may unevenly project to and/or activate the direct and indirect pathways of basal ganglia [19–21], which have been suggested or implied to be crucial for learning from positive and negative feedbacks, respectively [22–27] (Fig 1B), and their computational roles have been studied [22,28–32]. Different circuits for positive and negative learning other than these pathways have also been suggested [33,34]. Given these, it is conceivable that SR-based system and IR-based system differentially learn from positive and negative RPEs in the brain (Fig 1C). Recent work [35] explored this possibility, and found that combination of SR- and IR-based systems learning mainly from positives and negative RPEs, respectively (referred to as the appetitive SR + aversive IR agent), performed well in certain dynamic reward environments. This work further suggested that implementation of such a combination in the cortico-basal ganglia pathways seems in line with various anatomical and physiological findings, including activations indicative of SR in limbic or visual cortices [17,18] and preferential connections from limbic and visual cortices to the direct pathway of basal ganglia [20,21].

Crucially, SR has a similarity to the eligibility trace (c.f., its "forward view" [36]) while model-free learning without eligibility trace is naturally described by using IR. Given this correspondence, we expected that the appetitive SR + aversive IR agent could potentially show similar behavior to the one shown by the agent having long / short eligibility traces for positive / negative RPEs in the abovementioned recent work [10], in particular, enhancement of obsession-compulsion cycle. Moreover, we also expected that fitting of the appetitive SR + aversive IR agent's behavior could potentially result in smaller weights of model-based control than the case of the agent with SR-based system only in the two-stage decision task [8], whereas eligibility-trace imbalance *per se* would not bias the estimation of the degree of model-based control. Here we addressed these expectations. Psychologically, in the appetitive SR + aversive IR agent, reinforcing effects of the pleasantness of a relief from obsession (though not necessarily consciously perceived) operate more generally or widely (or in longer time scales due to the similarity between SR and eligibility trace) than reinforcing effects of the unpleasantness of a stay at obsession. In other words, this agent is characterized by (over)generalization of the pleasantness of a relief from obsession.

## Results

### Environmental model and dual-system agent

We adopted the previously developed environmental model that describes possible enhancement of obsession-compulsion cycle [10] (Fig 2). There are two states named the relief state and the obsession state. At the relief state, the agent can take one of two actions named the "abnormal reaction" and the "other" action. Here it is presumed that occasionally some anxiety e.g., about whether the door was locked arises spontaneously (intrusive thought), and the agent can either get deeply into the anxiety ("abnormal reaction"), entering the obsession state,

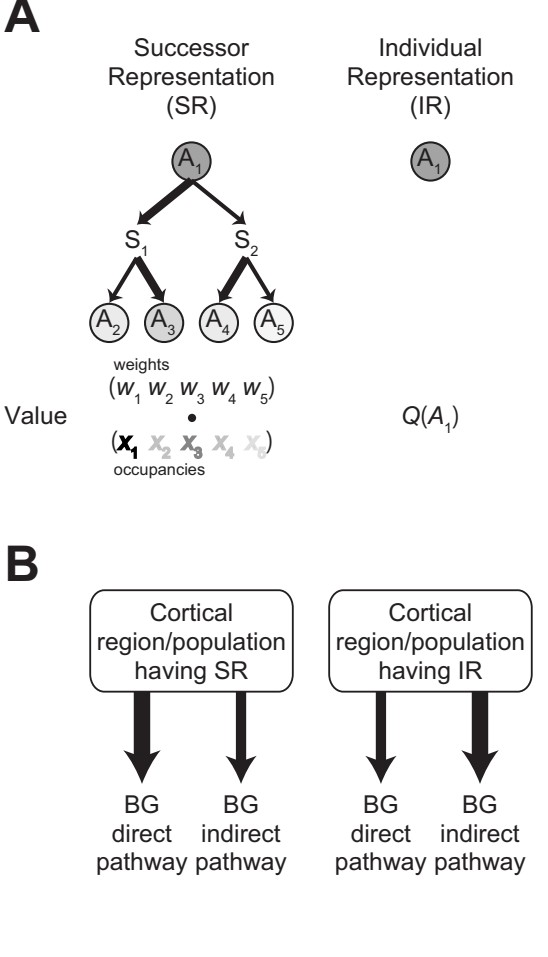

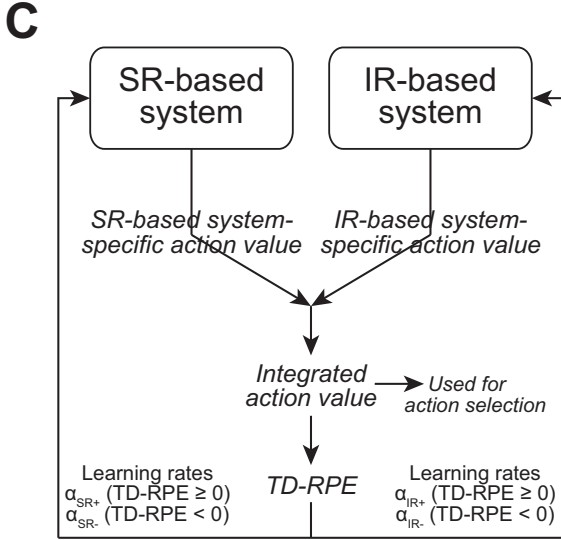

**Fig 1. The dual-system agent having a coupled successor-representation(SR)-based system and individual-representation(IR)-based system, adopted from the previous study [35]. (A)** Schematic illustrations of SR (left) and IR (right). In SR, action $A_1$ is represented by a set of "discounted cumulative occupancies" of its successor actions (including $A_1$ itself), i.e., (temporally discounted) cumulative frequencies with which each successor action is taken, starting from $A_1$, under a given policy in the environment. The widths of the arrows indicate the probabilities of state transitions (to state $S_1$ or $S_2$) and action selections ($A_2$ or $A_3$ at $S_1$ and $A_4$ or $A_5$ at $S_2$), and the darkness of each circle

indicates the occupancy of each action. Value of $A_1$ is given by the dot product of the vector of occupancy of each action $(x_1 \, x_2 \, x_3 \, x_4 \, x_5)$ and a weight vector $(w_1 \, w_2 \, w_3 \, w_4 \, w_5)$, which is updated by reward prediction errors (RPEs). By contrast, in IR, action $A_1$ is represented just by itself, separately from other actions. Value of $A_1$, $Q(A_1)$, is directly updated by RPEs. **(B)** Different cortical regions/populations having SR or IR may unevenly target/activate the direct and indirect pathways of basal ganglia (BG), which have been suggested to be crucial for learning from positive and negative feedbacks, respectively. The line widths of the arrows indicate the suggested preferences of projections/activations described in the Introduction and [35]. **(C)** The dual-system agent incorporating the potentially uneven projections/activations from the cortical regions/populations having SR or IR to the BG pathways. Each of the SR-based system and the IR-based system develops the system-specific value of each action, and the average of the two system-specific values, named the integrated action value, is used for soft-max action selection and calculation of SARSA-type TD-RPEs. The TD-RPEs are used for updating the system-specific values (or more specifically, the IR-based system-specific values and the weights for the SR-based system-specific values), with the learning rates for the SR- and IR-based systems in the cases of positive (non-negative) and negative TD-RPEs, denoted as $\alpha_{SR+}$, $\alpha_{SR-}$, $\alpha_{IR+}$, and $\alpha_{IR-}$, can take different values.

or just ignore or forget about it and do other things ("other" action), remaining in the relief state. Once the agent enters the obsession state, the agent again can take one of two actions. If the agent takes the "compulsion" action by paying a small cost, e.g., confirms door lock, the agent can transition back to the relief state with a high probability. Alternatively, the agent can take "other" action without any cost, but it makes the agent transition back to the relief state only with a small probability. Every stay at the obsession state imposes punishment (negative reward).

Because of this punishment and also the cost, the optimal value of the obsession state should be negative and lower than the optimal value of the relief state. Thus, the optimal value of the "abnormal reaction", which induces transition to the obsession state, should be negative and lower than the optimal value of the "other" action at the relief state. Therefore, normative reinforcement learning (RL) agents should learn a policy that minimizes selection of the "abnormal reaction", thereby never developing enhanced obsession-compulsion cycle. However, agents that deviate from normative RL, such as the one having different lengths of eligibility traces for positive versus negative RPEs examined in the previous work [10], may behave differently.

We examined how the dual-system agent having coupled SR- and IR-based systems (Fig 1C), originally developed in [35], behaved in this environmental model. As mentioned in the Introduction (and also in [35]), this dual-system agent was motivated by the suggestions that humans/animals use both model-based (SR-based) and model-free (IR-based) controls, which may learn differently from positive and negative RPEs through uneven projections from different cortical populations hosting different representations (SR or IR) to the direct and indirect pathways of basal ganglia. In this dual-system agent, each system develops the system-specific value of each action, and the average of the two system-specific values, named the (integrated) action value, is used for soft-max action selection and calculation of RPEs. The RPEs are used for updating the system-specific values, with the learning rates for the SR- and IR-based systems in the cases of positive and negative RPEs, denoted as $\alpha_{SR+}$, $\alpha_{SR-}$, $\alpha_{IR+}$, and $\alpha_{IR-}$, allowed to take different values.

## Behavior of the dual-system agent in the environmental model

First, we confirmed whether the agent that effectively had only the IR-based system with the same learning rates from positive and negative RPEs, i.e., the conventional model-free agent, behaved in the normative manner as described above, by setting the learning rates of the SR-based system from positive and negative RPEs ($\alpha_{SR+}$ and $\alpha_{SR-}$) to 0 while $\alpha_{IR+}$ and $\alpha_{IR-}$ were set to 0.1. Fig 3A-top shows an example of the time evolution of the difference in the (integrated) action values of "abnormal reaction" and "other" at the relief state ($Q_{abnormal} - Q_{other@relief}$). The difference became negative, indicating that the abnormal reaction became unlikely

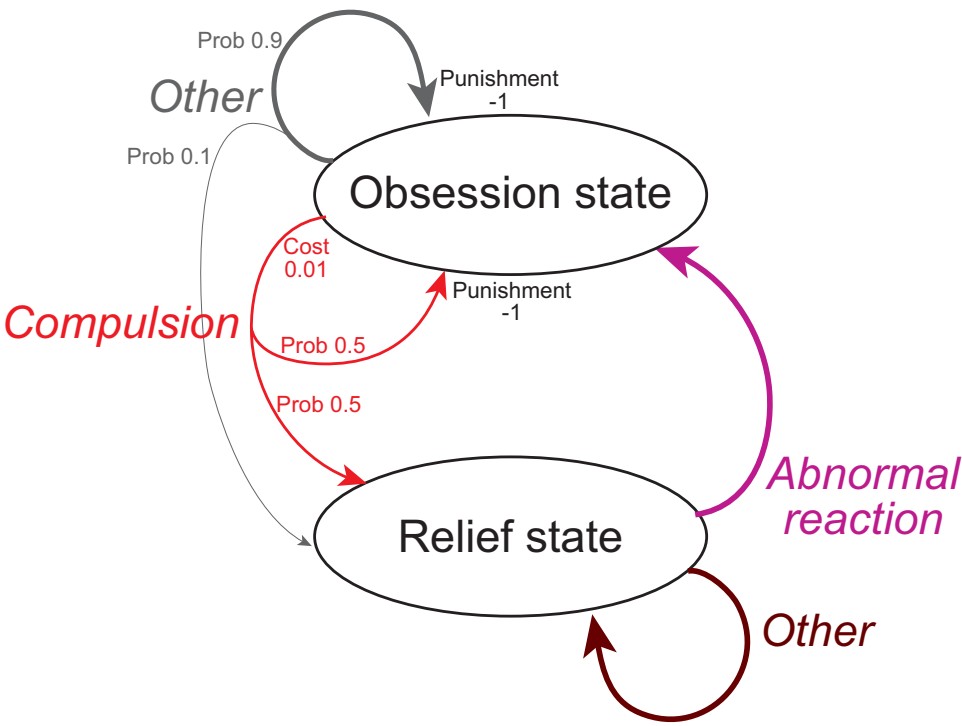

$$P(action) = \frac{\exp(\beta V(action))}{\sum_{available\ actions}\{\exp(\beta V(action))\}}$$

**Fig 2. Environmental model describing possible development of obsession-compulsion cycle, adopted from the previous study [10].** There are two states: the relief state and the obsession state. At the relief state, the agent can take the "abnormal reaction" (to an intrusive thought, i.e., spontaneously arising anxiety e.g., about door lock), which induces a transition to the obsession state, or the "other" action (e.g., just ignore or forget about the intrusive thought), with which the agent stays at the relief state. At the obsession state, the agent can take the "compulsion" action (e.g., confirms door lock), which requires a small cost (0.01) but induces a transition back to the relief state with a high probability (50%), or the "other" action, which requires no cost but induces a transition back to the relief state only with a small probability (10%). Every stay at the obsession state imposes punishment (negative reward −1).

to be taken. Fig 3A-middle shows the moving average of the proportion that the agent was at the obsession state in the same example simulation, and Fig 3A-bottom shows the average of such a moving-average proportion across 100 simulations. The proportion became, and remained to be, low. It was thus confirmed that this agent minimized selection of the

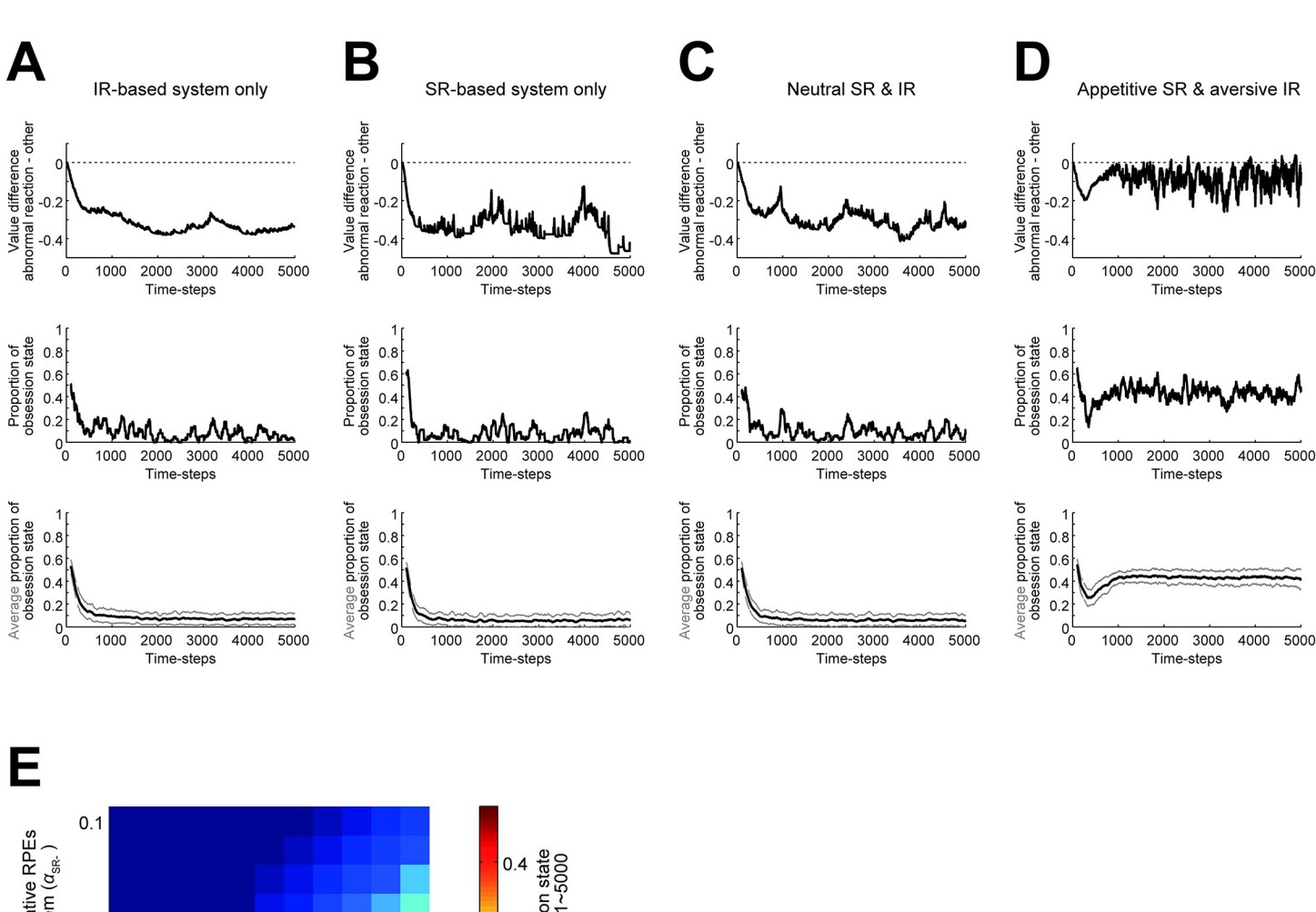

**Fig 3. Behavior of different types of the dual-system agents.** **(A)** Behavior of the agent that effectively had IR-based system only (($\alpha_{SR+}$, $\alpha_{SR-}$, $\alpha_{IR+}$, $\alpha_{IR-}$) = (0, 0, 0.1, 0.1)). *Top*: An example of the time evolution of the difference in the action values of the "abnormal reaction" and the "other" at the relief state ($Q_{abnormal} - Q_{other@relief}$). *Middle*: The moving average (over the past 100 time steps) of the proportion that the agent was at the obsession state in the same example simulation. *Bottom*: The average of the moving-average proportion of the obsession state across 100 simulations (black line), presented with ±SD (gray thin lines). **(B)** Behavior of the agent that effectively had SR-based system only (($\alpha_{SR+}$, $\alpha_{SR-}$, $\alpha_{IR+}$, $\alpha_{IR-}$) = (0.1, 0.1, 0, 0)). **(C)** Behavior of the agent having IR- and SR-based systems, both of which learned equally from positive and negative RPEs (($\alpha_{SR+}$, $\alpha_{SR-}$, $\alpha_{IR+}$, $\alpha_{IR-}$) = (0.05, 0.05, 0.05, 0.05)). **(D)** Behavior of the agent having appetitive SR- and aversive IR-based systems (($\alpha_{SR+}$, $\alpha_{SR-}$, $\alpha_{IR+}$, $\alpha_{IR-}$) = (0.09, 0.01, 0.01, 0.09)). **(E)** Proportion of the obsession state during time-steps 4901~5000, averaged across 100 simulations, in various cases with different learning rates. The horizontal and vertical axes indicate the learning rates of the SR-based system from positive and negative RPEs (i.e., $\alpha_{SR+}$ and $\alpha_{SR-}$), respectively, while $\alpha_{SR+} + \alpha_{IR+}$ and $\alpha_{SR-} + \alpha_{IR-}$ (i.e., total learning rates from positive and negative RPEs, respectively) were kept constant at 0.1.

"abnormal reaction" at the relief state and never developed enhanced obsession-compulsion cycle. We next examined the case where the agent effectively had only the SR-based system with unbiased learning from positive and negative PREs, by setting $\alpha_{IR+}$ and $\alpha_{IR-}$ to 0 while

$\alpha_{SR+}$ and $\alpha_{SR-}$ were set to 0.1. As shown in Fig 3B, this agent also minimized selection of the "abnormal reaction". This is actually reasonable given that SR-based temporal-difference (TD) leaning can generally approximate optimal value functions.

Next, we examined the behavior of the agent in which both IR-based system and SR-based system were effective and the learning rate was all equal for both systems and also for positive and negative RPEs (i.e., $\alpha_{IR+} = \alpha_{IR-} = \alpha_{SR+} = \alpha_{SR-} = 0.05$, referred to as the neutral SR+IR agent). As shown in Fig 3C, this agent also minimized selection of the "abnormal reaction". This may also be reasonable, as this agent combined rigorous normative model-free TD leaner and approximation of normative TD leaner. We then examined the behavior of the agent having appetitive SR- and aversive IR-based systems (referred to as the appetitive SR + aversive IR agent) by setting $\alpha_{SR+} = \alpha_{IR-} = 0.09$ and $\alpha_{SR-} = \alpha_{IR+} = 0.01$ (Fig 3D). As shown in Fig 3D-top, different from the previous cases, the $Q_{abnormal} - Q_{other@relief}$ difference did not smoothly decrease but remained to fluctuate and sometimes even became positive, indicating that this agent continued to take the "abnormal reaction" and transition to the obsession state rather frequently, as actually appeared in Fig 3D-middle,bottom. In other words, obsession-compulsion cycle was often developed and enhanced in this agent, as we expected. We further examined cases with different learning rates while $\alpha_{SR+} + \alpha_{IR+}$ and $\alpha_{SR-} + \alpha_{IR-}$ (i.e., total learning rates from positive and negative RPEs, respectively) were kept constant at 0.1, finding that agents with large $\alpha_{SR+}$ & small $\alpha_{IR+}$ and small $\alpha_{SR-}$ & large $\alpha_{IR-}$ (i.e., appetitive SR- and aversive IR-based systems) could develop enhanced obsession-compulsion cycle (Fig 3E).

## Modifications and variations in the models

We conducted simulations for longer times, and found that the enhanced obsession-compulsion cycle developed in the appetitive SR + aversive IR agent ultimately terminated (Fig 4A). We looked at the time evolution of variables, and found that the IR-system-specific values, as well as the weights for the SR-system-specific values (calculated as dot products of the weights and the occupancies: Fig 1A), grew to large magnitudes, while the magnitudes of the integrated action values and the SR matrix remained small (Fig 4B). This value/weight growth is considered to be due to the opponent setting rather than factors specific to SR, as value growth also occurred in the agent consisting of an appetitive IR-based system and an aversive IR-based system, for which obsession-compulsion cycle was not enhanced (Fig 4C). Returning to the appetitive SR + aversive IR agent, because the weights for the SR-system-specific values became so large, update of the SR matrix could cause large changes in the SR-system-specific values and thereby changes in the integrated values, whose impact could be quite large given the small magnitudes of the integrated values. This presumably triggered the termination of enhanced obsession-compulsion cycle. Indeed, if the rate of the update of SR matrix, which was so far fixed at 0.01, decreased over time and converged to 0, i.e., SR became rigid (c.f., [37]), enhanced obsession-compulsion cycle appeared to persist, despite the growth of the system-specific weights/values (Fig 4D and 4E). Considering neural implementation, however, such an unbounded growth of the weights/values may be implausible. As an alternative modification to the model, we introduced a small decay (forgetting) (c.f., [38,39]) to the original model with update of SR matrix at the fixed rate 0.01. Specifically, the IR-system-specific values and the weights for the SR-system-specific values were assumed to decay at a constant rate (0.001) per each time step. Then, the growth of these values and weights was bounded, and enhanced obsession-compulsion cycle persisted (Fig 4F–4H).

The reason for the stable persistence of enhanced obsession-compulsion cycle in the appetitive SR + aversive IR agent can be understood by looking at the action values and the SR matrix during the enhanced cycle (Fig 4G). The "other" action at the relief state had a positive value,

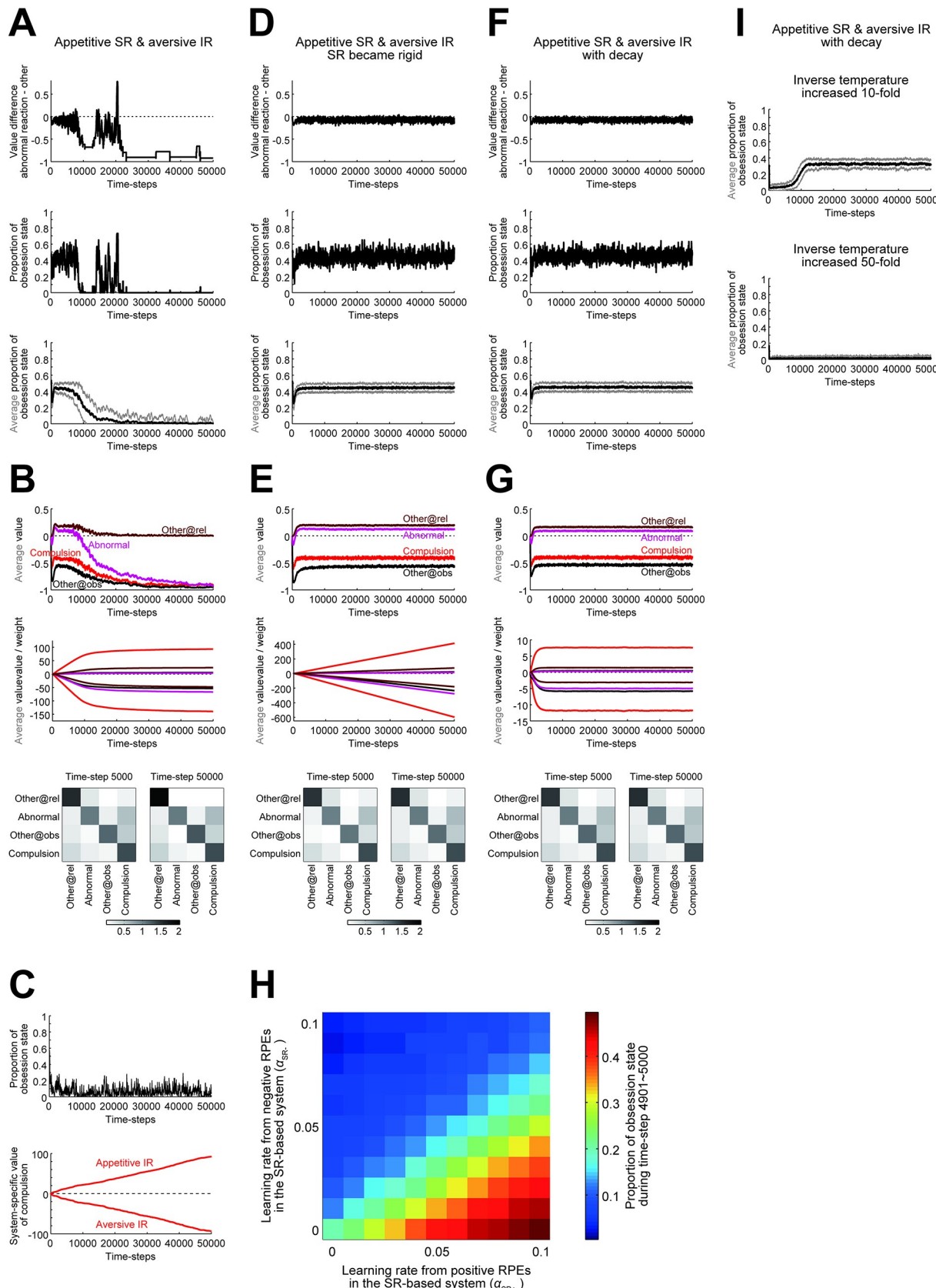

**Fig 4. Behavior of the original and modified dual-system agents in the long run. (A)** Behavior of the original appetitive SR + aversive IR agent for 50000 time steps. The learning rates were set as $(\alpha_{SR+}, \alpha_{SR-}, \alpha_{IR+}, \alpha_{IR-}) = (0.09, 0.01, 0.01, 0.09)$ (same for (A,B, D-F)). Arrangements are the same as those in Fig 3A–3D. **(B)** Integrated action values, system-specific values/weights, and SR matrix of the original appetitive SR + aversive IR agent. *Top*: Integrated value of each action averaged across 100 simulations (brown: "other" at the relief state, purple: "abnormal reaction", black: "other" at the obsession state, red: "compulsion"; same below). *Middle*: Weights for SR-based system-specific values (top four lines) and IR-based system-specific values (bottom four lines) averaged across 100 simulations. *Bottom*: SR matrix at time step 5000 (left) and 50000 (right) averaged across 100 simulations. Each row indicates the SR of each action shown in the left, with the darkness of the squares indicating the discounted cumulative occupancies of the actions shown in the bottom. **(C)** Behavior of the appetitive IR $((\alpha_{IR+}, \alpha_{IR-}) = (0.09, 0.01))$ + aversive IR $((\alpha_{IR+}, \alpha_{IR-}) = (0.01, 0.09))$ agent. The moving average of the proportion that the agent was at the obsession state (top) and the system-specific values of the compulsion (bottom) in a single simulation are shown. **(D,E)** Results for the appetitive SR + aversive IR agent with a modification, in which the rate of the update of SR matrix decreased over time according to 0.01/(1 + time-step/1000). **(F,G)** Results for the appetitive SR + aversive IR agent with another modification, in which the IR-based system-specific values and the weights for the SR-based system-specific values decayed at a constant rate (0.001 at each time step) while the rate of the update of SR matrix was fixed at 0.01 as in the original agent. **(H)** Average proportion of the obsession state during time-steps 49901~50000 for the modified agent with the decay of values/weights, in various cases with different learning rates. Notations are the same as those in Fig 3E. **(I)** The average of the moving-average proportion of the obsession state across 100 simulations (black line), presented with ±SD (gray thin lines), of the appetitive SR + aversive IR agent with the decay of values/weights, with the inverse temperature increased tenfold (top) or fifty-fold (bottom: −SD is mostly invisible).

and the abnormal reaction had a lower but still positive value. These positive values are considered to be shaped through asymmetric SR-based updates by RPEs generated when the agent took an action at the obsession state. Specifically, positive RPE upon transition to the relief state had large effects since $\alpha_{SR+}$ was large whereas negative RPE upon stay at the obsession state had small effects since $\alpha_{SR-}$ was small. Now, assume that the agent, developing enhanced obsession-compulsion cycle, exists at the relief state and happens to take the "other" action repeatedly, without taking the abnormal reaction. It is in fact the optimal policy, and because no reward or punishment is given as long as the agent only takes the "other" action, the value of the "other" action should approach, through RPE-based update, to 0, which is equal to the optimal value. However, this value, 0, is lower than the positive value of the abnormal reaction (even though it slowly decays). Therefore, as the agent repeatedly takes the "other" action more and more, ironically the abnormal reaction becomes more and more likely to be chosen. This pulls the agent back to enhanced obsession-compulsion cycle, away from the optimal policy. Notably, because the value of the abnormal reaction was (positive but) lower than the value of the "other" action at the relief state as mentioned above, it seems possible that if the agent's choice policy is nearly deterministic (i.e., the inverse temperature is quite high), the agent would rarely take the abnormal reaction and thereby rarely develop enhanced obsession-compulsion cycle. We examined this possibility. When the inverse temperature increased tenfold (from $\beta = 10$ to $\beta = 100$), development of enhanced obsession-compulsion cycle was on average fairly delayed (Fig 4I, top), and when it further increased fivefold ($\beta = 500$), enhanced cycle was on average not developed within the examined time steps (50000) (Fig 4I, bottom). These results support the abovementioned conjecture about the mechanism and suggest a possible effect of choice exploration/exploitation.

In the environmental model used so far (developed by [10]) (Fig 2), there was only a single "other" action at each state. It is conceivable to assume multiple (different) "other" actions (Fig 5A). We examined the behavior of the dual-system agent with decay in modified environments where there were two or eight "other" actions at each state. When there were two "other" actions at each state (Fig 5B and 5C), the appetitive SR + aversive IR agent still exhibited enhanced obsession-compulsion cycle, although the proportion of the obsession state decreased from the original case with a single "other" action. The value of the abnormal reaction in such an agent was around 0 (purple line in Fig 5B middle), and so the abovementioned mechanism for enhanced obsession-compulsion cycle would barely hold. However, when there were eight "other" actions at each state (Fig 5D and 5E), the value of the abnormal reaction became negative (purple line in Fig 5D middle) and obsession-compulsion cycle was

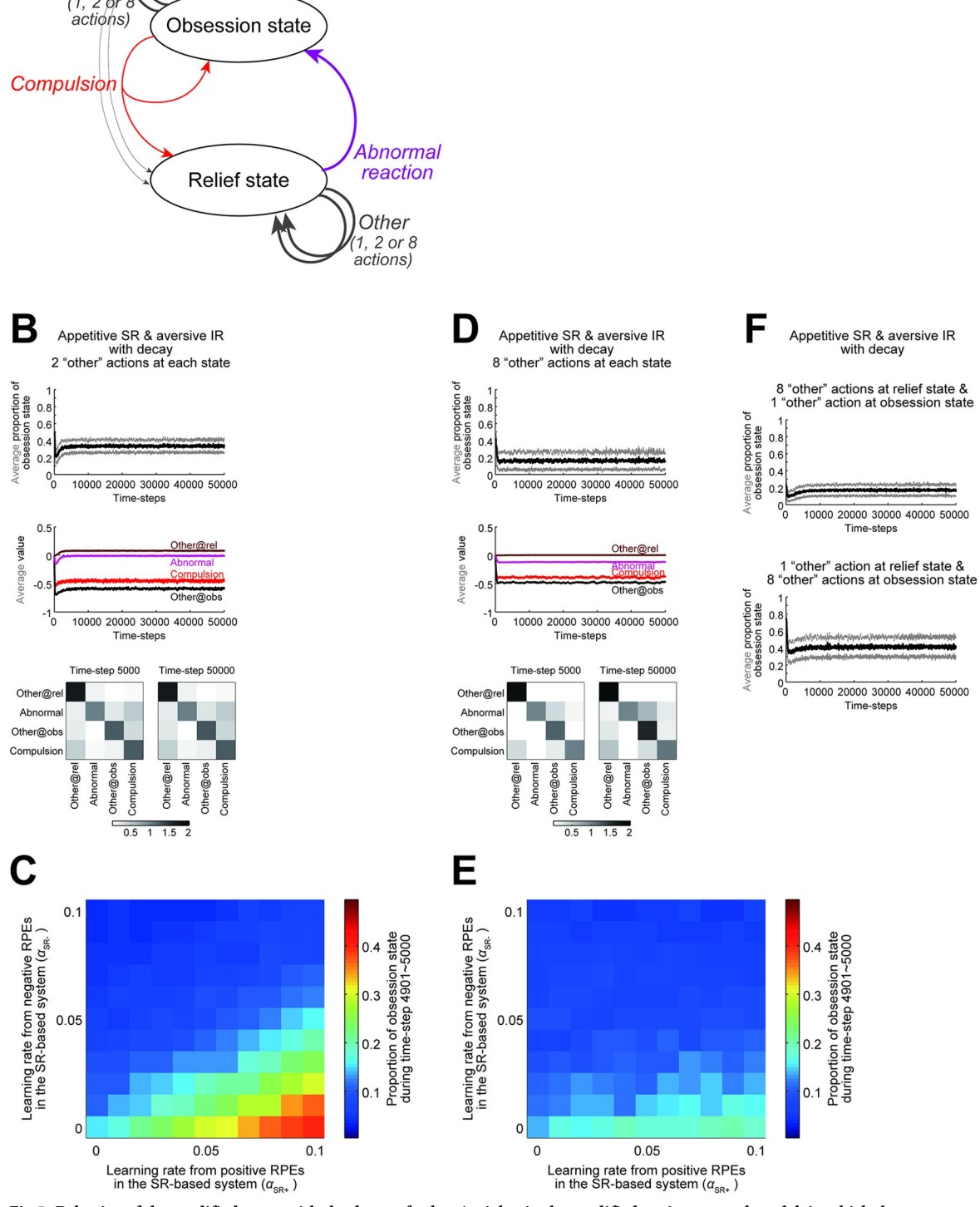

**Fig 5. Behavior of the modified agent with the decay of values/weights in the modified environmental model, in which there were multiple "other" actions. (A)** Modified environmental model assuming multiple "other" actions. **(B)** Behavior of the appetitive SR + aversive IR agent ($(\alpha_{SR+}, \alpha_{SR-}, \alpha_{IR+}, \alpha_{IR-}) = (0.09, 0.01, 0.01, 0.09)$) with the decay of values/weights in the environment with two "other" actions at each state. *Top*: The average of the moving-average proportion of the obsession state across 100 simulations (black line), presented with ±SD (gray thin lines). *Middle*: Integrated action value averaged across 100 simulations (brown: average of "other" actions at

the relief state, purple: "abnormal reaction", black: average of "other" actions at the obsession state, red: "compulsion"). *Bottom*: SR matrices averaged across 100 simulations, in which each row indicates the SR of "abnormal reaction" and "compulsion" and the mean SR of "other" actions at each state shown in the left (i.e., averaged across "other" actions at the same states), with the darkness of the squares indicating the discounted cumulative occupancies of "abnormal reaction" and "compulsion" and the summed discounted cumulative occupancies of "other" actions at each state shown in the bottom (i.e., summed across "other" actions at the same states). **(C)** Average proportion of the obsession state during time-steps 49901~50000 for the agent with the decay of values/weights, in various cases with different learning rates, in the environment with two "other" actions at each state. The color bar is the same as the one in Fig 4H. **(D,E)** Results for the cases where there were eight "other" actions at each state in the environmental model. **(F)** Behavior (average ±SD of the moving-average proportion of the obsession state) of the appetitive SR + aversive IR agent with the decay of values/weights in the environment where there were eight "other" actions at the relief state and one "other" action at the obsession state (top) or one "other" action at the relief state and eight "other" actions at the obsession state (bottom).

hardly enhanced even in the appetitive SR + aversive IR agent. We further examined the cases where there were eight "other" actions at the relief state and one "other" action at the obsession state (Fig 5F, top) or one "other" action at the relief state and eight "other" actions at the obsession state (Fig 5F, bottom). Enhanced obsession-compulsion cycle was developed only in the latter case. These results suggest that the reduced representation of actions other than the abnormal reaction at the relief state in the original environmental model is essential for enhancement of obsession-compulsion cycle.

## Behavior of the dual-system agent in the two-stage decision task

We examined how the appetitive SR + aversive IR agent behaved in the two-stage decision task [8], in comparison to the agent with SR-based system only (referred to as the SR-only agent) that would correspond to healthy controls who had long memory traces for both positive and negative RPEs in the previous work [10]. Specifically, we simulated choice sequences of both agents 100 times for each using different pseudorandom numbers, and fitted them by the RL model weighing model-based and model-free control developed in the original study [8] to estimate the parameter $w$ that represents the degree of model-based control. As shown in the left panels of Fig 6A and 6B, the estimated $w$ for the appetitive SR + aversive IR agent (Fig 6B) was generally smaller (as a distribution) than that for the SR-only agent (Fig 6A) (Wilcoxon rank sum test, $p = 1.04 \times 10^{-7}$). This reduction of $w$ was rather prominent, even though the appetitive SR + aversive IR agent was half made of SR learner, and we think it could thus potentially explain the reduction of $w$ observed in OCD patients [7]. Notably, the distribution of the estimated $w$ for SR-only agent was not very close to 1 but rather wide, and this might reflect the fact that our SR-based system incorporated PE-based updates rather than direct value calculation by multiplication of fixed SR features and the reward vector. In contrast, fitting of choices generated by the original RL model with balanced eligibility trace ($\lambda = 0.4$ for both positive and negative RPEs) and those generated by a modified model with imbalanced trace ($\lambda = 0.4$ and 0 for positive and negative RPEs, respectively), both of which had the same degree of model-based control ($w = 0.5$), did not result in smaller estimated $w$ in the latter, as shown in the left panels of Fig 6C and 6D (Wilcoxon rank sum test, $p = 0.5134$). Therefore, our expectation was again verified, and this result indicates a possibility that the OCD patients who were suggested to have imbalanced eligibility trace [10] might not actually have imbalanced trace but instead have opponent SR+IR structure and overlap with previously examined OCD populations who showed impairment of model-based control [7].

We further conducted the fitting analysis for the neutral SR + IR agent. As shown in the left panel of Fig 6E, the estimated $w$ was typically low, and its distribution did not drastically differ from the case of the appetitive SR + aversive IR agent, although there was a tendency (Wilcoxon rank sum test, $p = 0.05857$). Thus, from the estimation of the weight of model-based control ($w$) only, we cannot say that the appetitive SR + aversive IR agent is a better model of

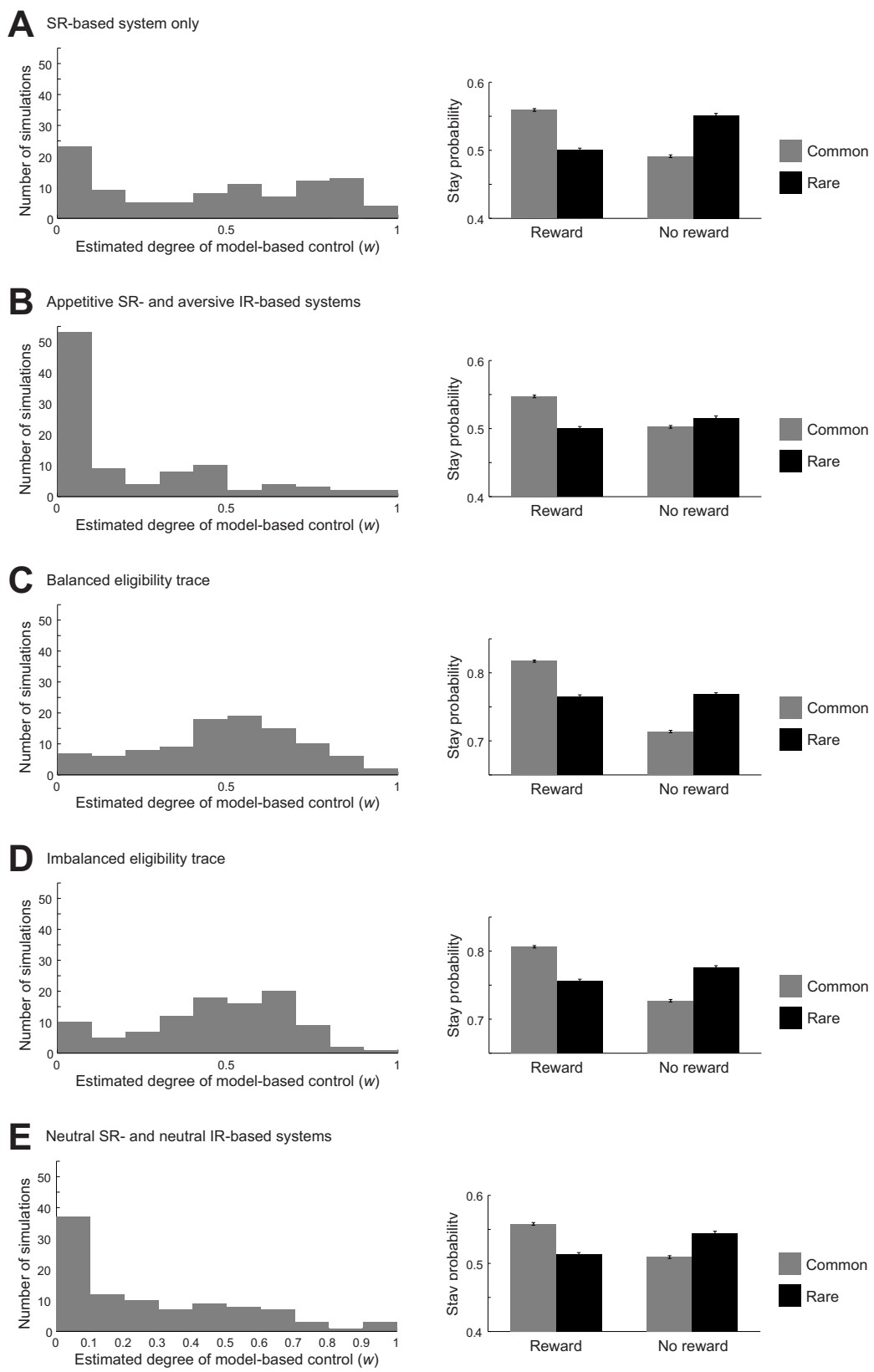

**Fig 6. Behavior of the dual-system agents in the two-stage decision task, as compared to the agent with balanced or imbalanced memory trace. (A,B)** Comparison between the SR-only agent (($\alpha_{SR+}$, $\alpha_{SR-}$, $\alpha_{IR+}$, $\alpha_{IR-}$) = (0.3, 0.3, 0, 0)) (A) and the appetitive SR + aversive IR agent (($\alpha_{SR+}$, $\alpha_{SR-}$, $\alpha_{IR+}$, $\alpha_{IR-}$) = (0.27, 0.03, 0.03, 0.27)) (B). *Left panels*: Distributions of the estimated parameter representing the degree of model-based control (*w*) across 97 simulations (out of 100 simulations: see the Materials and Methods for details). *Right panels*: Proportion of stay choice (i.e., choosing the same option as the one at the previous trial) at the first stage depending on whether reward was obtained or not and whether common or rare transition was occurred at the previous trial. The bars indicate the average across 1000 simulations, and the error-bars indicate ±SEM. **(C,D)** Comparison between the RL model developed in the original two-stage task study [8] with balanced eligibility trace ($\lambda = 0.4$) (C) and a modified model with imbalanced eligibility trace (($\lambda_1$, $\lambda_2$) = (0.4, 0) for positive and negative TD-RPE, respectively) (D). The degree of model-based control (*w*) was set to 0.5 in the simulations of both models. *Left panels*: Distributions of the estimated *w* across 100 simulations. *Right panels*: Mean (±SEM) proportion of stay choice at the first stage across 1000 simulations. **(E)** Results for the neutral SR + IR agent (($\alpha_{SR+}$, $\alpha_{SR-}$, $\alpha_{IR+}$, $\alpha_{IR-}$) = (0.15, 0.15, 0.15, 0.15)). Configurations are the same those in (A-D), with the left panel showing the distribution of *w* across 97 out of 100 simulations.

OCD patients than the neutral SR + IR agent, while such a claim was supported by the above-mentioned simulation results (Figs 3 and 4H).

We analyzed the choice pattern of each type of agent, focusing on how much the agent's choices (apparently) took into account whether the occurred state transition was the common type or the rare type. Specifically, we focused on the degree with which the agent made more "stay" (i.e., chose the same first-stage option) after reward was obtained in the case of common transition as compared to rare transition, and also made less stays after no reward was obtained in the case of common, as compared to rare, transition. The appetitive SR + aversive IR agent showed a smaller common-vs-rare difference in the "stay" proportion after no-reward than after reward (Fig 6B, right panel) (comparison of $(P(\text{Stay})|_{\text{common}} - P(\text{Stay})|_{\text{rare}})|_{\text{reward}}$ and $(P(\text{Stay})|_{\text{rare}} - P(\text{Stay})|_{\text{common}})|_{\text{no-reward}}$, $t(999) = 6.97$, $p = 5.93 \times 10^{-12}$), as naturally expected from the combination of appetitive/aversive learning and SR/IR. In contrast, in the SR-only agent, the neutral SR+IR agent, as well as the agents with balanced or imbalanced eligibility trace, the common-vs-rare difference in the "stay" proportion was similar between after reward and after no-reward (Fig 6A, 6C, 6D and 6E right panels). Therefore, whether the appetitive SR + aversive IR agent or the neutral SR+IR agent better describes OCD could be distinguished by examining whether there exists such an asymmetry between after reward and after no-reward in the common-vs-rare difference in the "stay" proportion.

## Behavior of the dual-system agents in the delayed feedback task

We examined whether the observed differential behaviors of OCD patients and HCs in a delayed feedback task [10] could be explained by our dual-system agents. In the task, a pair of stimuli were presented, and subjects were required to choose one of them. Some stimuli caused immediate positive or negative feedback (monetary gain or loss) in the same trial, while other stimuli caused delayed feedback three trials later. Compared to HCs, OCD patients showed intact learning of stimuli causing immediate feedback but impaired learning of stimuli with delayed feedback (Fig 3B, 3C of [10]), and this pattern was reproduced by models with balanced and imbalanced eligibility traces (Fig 3E, 3F of [10]). We simulated execution of this task by the dual-system agents, in which actions of choosing each stimulus were represented by estimated discounted cumulative occupancies of successor states defined based on presented feedback (see the Materials and Methods for details). As shown in Fig 7A and 7B, the appetitive SR + aversive IR agent ($\alpha_{SR+}$, $\alpha_{SR-}$, $\alpha_{IR+}$, $\alpha_{IR-}$) = (0.018, 0.002, 0.002, 0.018), as compared to the SR-only agent ($\alpha_{SR+}$, $\alpha_{SR-}$, $\alpha_{IR+}$, $\alpha_{IR-}$) = (0.02, 0.02, 0, 0), showed a particular impairment in learning of delayed feedback, largely reproducing the pattern observed in the experiment [10].

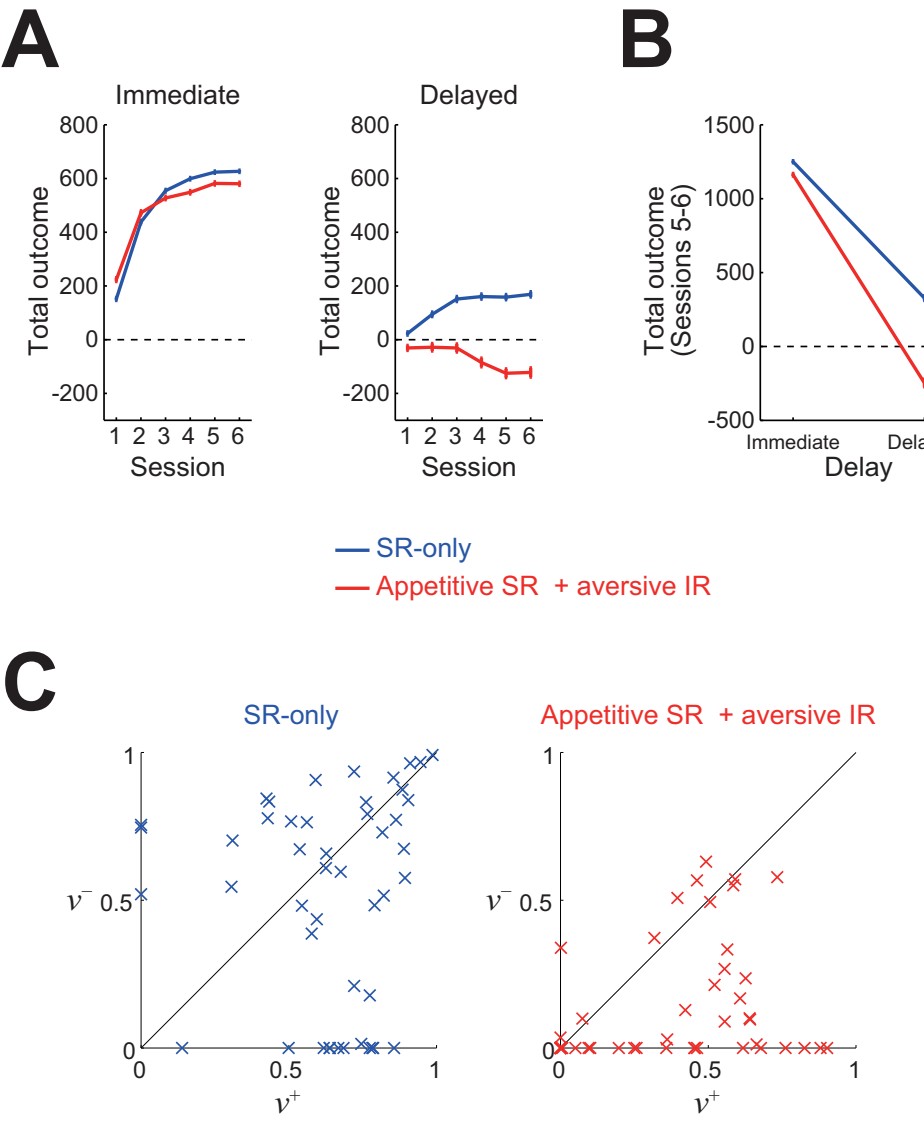

**Fig 7. Behavior of the dual-system agents in the delayed feedback task examined in [10]. (A)** Total obtained outcome (feedback) from stimuli causing immediate (left) or delayed feedback (right) in each session, averaged across 1000 simulations, for the appetitive SR + aversive IR agent (red) and the SR-only agent (blue). The error-bars indicate ±SEM. **(B)** Total obtained outcome (feedback) in sessions 5 and 6, averaged across 1000 simulations, from stimuli with immediate or delayed feedback for the two types of agents. The error-bars indicate ±SEM. **(C)** Results of fitting of the choices of the two types of agents (47 and 45 out of 50 simulations for each, separately conducted from those shown in (B)) by the separate eligibility trace actor-critic model considered in [10]. The horizontal and vertical axes indicate the estimated parameters $v^+$ and $v^-$ (decay time scale of the trace for positive and negative RPEs), respectively.

We further fitted the choices of these two types of dual-system agents by the separate eligibility trace actor-critic model considered in [10]. As a result, the choices of the SR-only agent were in large part fitted by the model with large eligibility traces for both positive and negative RPEs whereas the choices of the appetitive SR + aversive IR agent were in large part fitted by the model with a larger trace for positive than negative RPEs (Fig 7C), as we expected from the similarity between the eligibility trace and SR. These results indicate that the observed behaviors of OCD patients and HCs in the delayed feedback task [10] could be explained by the appetitive SR + aversive IR agent and the SR-only agent, respectively.

## Discussion

We have verified our two expectations: i) the appetitive SR + aversive IR agent could develop enhanced obsession-compulsion cycle, similarly to the agent having long / short eligibility traces for positive / negative RPEs examined in [10], and ii) fitting of the appetitive SR + aversive IR agent's behavior could result in smaller weights of model-based control than the SR-only agent in the two-stage decision task whereas eligibility-trace imbalance *per se* would not bias the estimation of the degree of model-based control. We have also shown that the appetitive SR + aversive IR agent and the SR-only agent could explain the behaviors of OCD patients and HCs, respectively, in the delayed feedback task in [10]. These results reconcile the recent suggestion of memory trace imbalance in OCD [10] with the long-standing suggestion that OCD is associated with impairment of model-based (goal-directed) control [1, 5, 7, 9], raising a possibility that opponent learning in model-based (SR-based) and model-free (IR-based) controls underlies obsession-compulsion. Below we discuss implications, limitations, and predictions of the present study, and also an alternative environmental model.

### Implications of our results, in relation to other studies

As described in the Introduction, the opponent combination of appetitive SR- and aversive IR-based systems has recently been shown to perform well in certain dynamic environments [35]. Specifically, in a virtual reward navigation task in which reward location dynamically changed, the appetitive SR + aversive IR agent outperformed the agents with other combinations. Its presumable reasons include a compensation of a potential weakness of SR-based learning from negative RPEs, indicated from the difficulty in learning using successor features (an extension of SR) upon drastic changes in the goal/policy [40], by IR-based aversive learning. Moreover, implementation of such an appetitive SR & aversive IR combination in the cortico-basal ganglia pathways appears consistent with various physiological and anatomical findings (c.f., Fig 11 of [35]), including activations indicative of SR in limbic or visual cortices [17,18] and preferential connections from limbic and visual cortices to the direct pathway of basal ganglia [20,21]. These previous suggestions, together with the results of the present study that the opponent SR+IR combination could develop enhanced obsession-compulsion cycle, potentially explain a mystery shown in the recent work [10]. Specifically, in that work, fitting of even HCs tended to result in shorter memory trace for negative (than positive) RPEs, and the authors suggested that it could indicate humans' proneness to OCD, but why humans may have such a bias in the first place remained unexplained. We propose that the opponent SR+IR combination has been evolutionarily selected because it is advantageous under certain dynamic environments, at the cost of proneness to OCD.

We have shown that when the number of possible actions other than the abnormal reaction at the relief state was increased, obsession-compulsion cycle was not enhanced. This result suggests that a factor contributing to enhanced obsession-compulsion cycle is that alternative actions are represented in a dimension-reduced manner. Intuitively, this could correspond to a situation where a person concentrates on the emerging intrusive thoughts so much that s/he cannot think of a variety of other things. Previous modeling work [37] suggested that dimension-reduced SR could relate to addiction if it is rigid, i.e., not updated according to changes in the policy. Empirical work [41] has shown that compulsivity is associated with impairment of state transition learning. In the present work, rigidness of SR was needed for stable retention of enhanced obsession-compulsion cycle in the original model without decay, although not for the modified model with decay. Taken together, dimension-reduction and rigidness of state/action representation could generally relate to psychiatric disorders/symptoms.

Our model does not specifically describe how the SR- and IR-based systems are arbitrated and how they are modulated by medical treatments. Serotonin has been suggested to be related

to arbitration of model-based and model-free control [7,42], and recent work demonstrated that silencing of dorsal raphe serotonin neurons disrupted model-based control in outcome devaluation tasks [43]. Meanwhile, selective serotonin reuptake inhibitors (SSRIs) have been used for treatment of OCD, and one of the suggested mechanisms of its effectiveness is that SSRIs re-engage the brain system for model-based control [1]. Our modeling results indicate a possibility that OCD is linked to impairment of learning of SR-based (model-based) system particularly from negative RPEs, which could be caused by serotonin depletion and recovered by SSRIs. This possibility is potentially consistent with a previous result shown in [42] that diminishing serotonin signaling by tryptophan depletion appeared to impair model-based choice after no-reward but not after reward in the two-stage decision task (their Fig 2A, the left two panels), similarly to the pattern predicted by our model (Fig 6B right panel).

We modeled the behavior of healthy subjects in the two-stage task by the SR-only agent rather than by the neutral SR+IR agent. This could be justified by the results that the choices of the neutral SR+IR agent were fitted with rather low weights of model-based control (*w*) while those of the SR-only agent were fitted by relatively large values of *w* (Fig 6A and 6E, left panels). We consider, however, that healthy people would not always use SR(model)-based learning but may also use IR-based learning, or even appetitive SR- and aversive IR-based learning when it is advantageous. Healthy people might be able to strongly rely on SR(model)-based system when appropriate while OCD patients might have an impairment in such an arbitration of behavioral control, potentially in line with a previous suggestion [44].

Previous studies using probabilistic reversal learning tasks reported unchanged / increased learning rate for positive / negative RPEs in OCD patients [45] or increased / decreased learning rate for positive / negative RPEs in youth with OCD [46], and a recent study using a probabilistic instrumental learning task reported unchanged / decreased learning rate for positive / negative RPEs in OCD patients [47]. Reasons for these mixed results remain elusive, but our appetitive SR + aversive IR agent (large / small learning rate for positive / negative RPE in the SR-based system but the opposite pattern in the IR-based system) could potentially explain them. Moreover, the recent study [47] has further shown that the decreased learning rate for negative RPEs in OCD was associated with attenuated representation of negative RPEs in the dorsomedial prefrontal cortex and the caudate. Given that the caudate (like dorsomedial striatum in rodents) has been implicated in model-based control (while putamen/dorsolateral striatum has been implicated in model-free control) [4,48], the attenuated representation of negative RPEs in the caudate in OCD can be consistent with our appetitive SR-based system having smaller learning rate for negative RPEs.

## Limitations

The previous studies examining the two-stage decision task in OCD subjects [7] or subjects with tryptophan depletion [42] also examined a different, "punishment version" of the two-stage task, in which reward or no-reward outcomes in the original task were replaced with no-punishment or punishment outcomes. Different from the results in the reward version of the task, in the punishment version, OCD subjects did not show impaired model-based choice [7] and tryptophan depletion rather promoted model-based choice [42]. The environmental model that we used (developed by [10]) only contained punishment and cost, and so it can be said to have a similarity to the punishment version, rather than the reward version. Nonetheless, even if the environmental model was altered in a way that punishment (−1) for the stay at the obsession state was replaced with no reward for the stay at the obsession state and reward (1) for all the other cases, the behavior of the dual-system agent without decay was largely preserved provided that the IR-system-specific values and the weights for SR-system-specific

values were initialized to $1+\gamma+\gamma^2+\ldots = 1/(1-\gamma) = 2$ instead of 0. But still, the lack of outcome-valence dependence is an important limitation of our model (we will discuss an alternative environmental model below). It was suggested [42] that serotonin's possible role in representation of the average reward rate [49] may be related to such outcome valence-dependent differential effects of tryptophan depletion. Exploration of how our model can be integrated into the average reward RL framework (c.f., [49]) is a possible future direction.

Our modeling results also appear not consistent with the result of a study examining generalization in OCD patients [50]. Using sensory preconditioning paradigm with monetary rewards and losses, this study reported that OCD patients generalized less from rewards, but not losses, than heathy controls (HCs). Looking at their results, in fact the HCs did not on average show significant generalization from rewards but showed chance-level choices, and the OCD patients significantly avoided the stimulus associated with a second stimulus that was later associated with stochastic (risky) gain and preferred the alternative stimulus associated with a second stimulus later associated with neutral outcome (Fig 3A of [50]). In the second-stimulus-outcome (gain/loss/neutral) association phase of the task, participants were required to "bid" on the presented stimulus and received feedback regardless of their bidding decisions, whereas no feedback was given in the final choice phase of the task. Given these task properties and observed choice patterns, we speculate that the task worked as a test for generalization of aversiveness associated with riskiness and/or loss of opportunity that arose when participants did not "bid" on reward-associated stimulus, and the OCD patients showed such a generalization of aversiveness while the HCs did not. As SR-based learning is more generalizable than IR-based one, the appetitive SR & aversive IR combination of our model predicts lager generalization from positive than negative RPEs in OCD, apparently opposite to the abovementioned conjecture.

However, this contradiction could potentially be resolved by considering recent findings of dopamine neuronal subpopulation representing threat prediction errors (PEs) rather than RPEs and striatal subpopulations engaged in threat (rather than reward) learning/management, including the tail of the striatum (TS) and substantia nigra pars lateralis (SNL) in mice [51–53], and also dopamine neuronal subpopulations potentially representing error signals for aversive outcome [54] or reward-omission [55]. Specifically, it is conceivable that greater learning of SR-based system from positive than negative dopamine signals in OCD suggested by our model occurs not only in the canonical cortico-basal ganglia-dopamine circuit for reward RL but also in the circuit for RL of threat or aversiveness, potentially through common mechanisms for corticostriatal projection preferences. Then, the latter circuit, which is recruited in punishment contexts, should implement the aversive SR + appetitive IR agent (Fig 8) and show (over)generalization of threat or aversiveness, potentially in line with the above-mentioned conjecture. Existence of such multiple opponent SR+IR circuits could further explain the previous mixed results regarding learning rate biases in OCD discussed above.

From a more general viewpoint, a potential limitation of our model is that it may not explain a wide range of phenomena associated with OCD. In particular, it has been suggested that OCD is accompanied with various degrees of obsession and compulsion, and there can be pure compulsion (compulsive hoarding can be it [1]) and pure obsession [56] (but see [1]). By contrast, the environmental model that we used, developed by [10], assumes the obsession-compulsion cycle as an inseparable core of OCD. The model in fact assumes that intrusive thoughts universally occur even in healthy people, and what distinguishes OCD is the "abnormal reaction" to such thoughts. Once entering the obsession state, even the SR-only agent is likely to choose the "compulsion" as it should have a larger value than the "other" because the "other" leads to punishment more frequently than the "compulsion". So in this model, OCD is not characterized by an increase in the value/choice-frequency of compulsion. The model

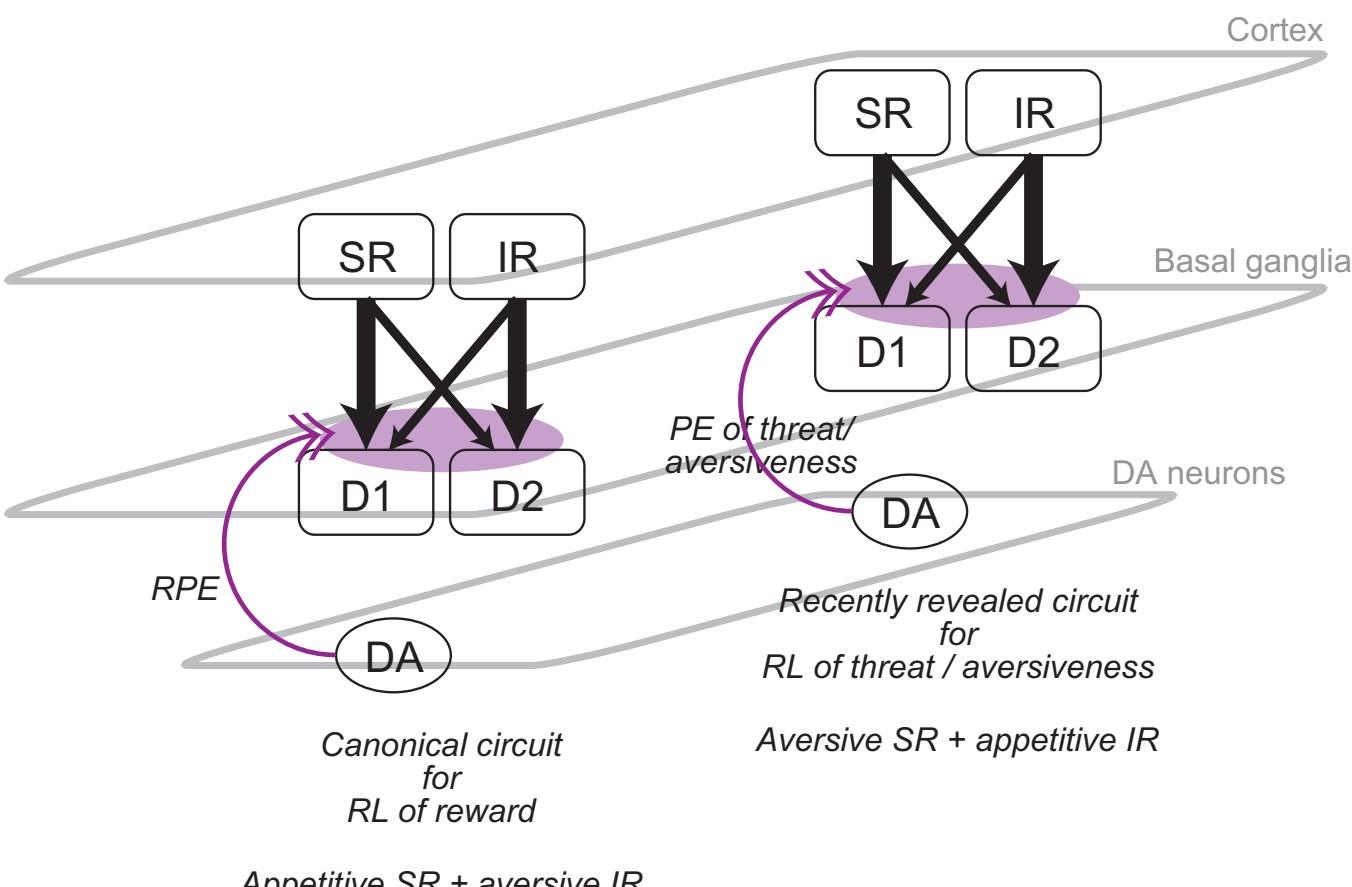

**Fig 8. Schematic diagram of the proposed opponent SR + IR learning in multiple parallel cortico-basal ganglia (BG)-dopamine (DA) circuits.** It is hypothesized that there exist preferential connections from the cortical populations having SR and IR to the striatal direct and indirect pathway neurons expressing D1 and D2 receptors, respectively. Such preferential connections cause greater learning of SR- and IR-based systems from positive and negative DA signals, respectively. It implements the appetitive SR + aversive IR agent in the canonical cortico-BG-DA circuit for reward reinforcement learning (RL), in which DA represents reward prediction error (RPE) (left part of this figure). In contrast, in the recently revealed cortico-BG-DA circuit for threat/aversiveness RL, in which DA represents threat/aversiveness PE, the same greater learning of SR/IR-based systems from positive/negative DA signals implements the aversive SR + appetitive IR agent (right part of this figure).

could still be compatible with the existence of (apparently) pure compulsion if the "abnormal reaction/obsession" can be subconscious, and also with pure obsession if the "compulsion" is not a motor action but a thought (as implied in [1]), presumably depending on which part of cortico-basal ganglia circuits has the opponent SR+IR structure. However, it has also been suggested that compulsion (rather than obsession or intrusive thoughts) is a key factor of OCD [1] and obsession can emerge from compulsion through erroneous reverse inference [57]. Future studies are desired to test these different theories, or possibly integrate them by elaborating the environmental model, which is currently rather simple (we will discuss an alternative environmental model below). Given that environmental complexity was suggested [58] to have crucial effects in addiction, which also entails compulsion, future elaboration of models and exploration of parameters are highly desired.

## Predictions, and comparisons with actual data

Our model predicts that OCD patients show smaller common-vs-rare differences in the "stay" proportion after no-reward than after reward in the two-stage decision task (Fig 6B, right

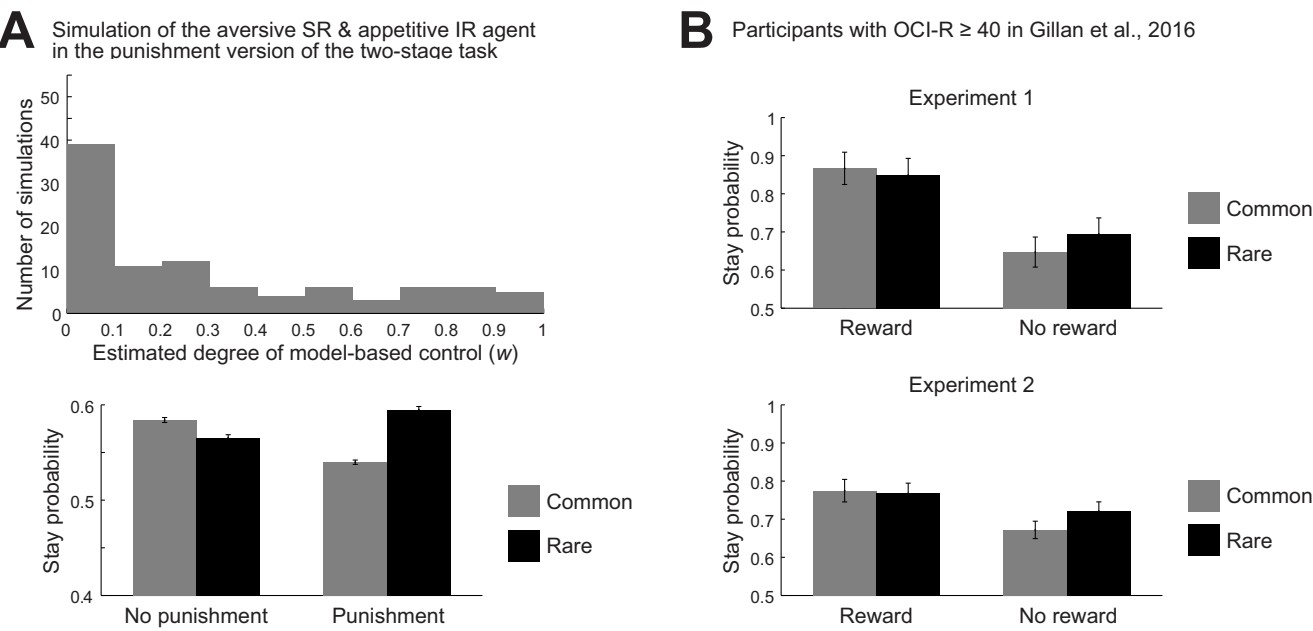

**Fig 9. Behavior in the two-stage decision task: additional simulation result and results obtained by analysis of publicly available experimental data.**
**(A)** Behavior of the aversive SR + appetitive IR agent $((\alpha_{SR+}, \alpha_{SR-}, \alpha_{IR+}, \alpha_{IR-}) = (0.03, 0.27, 0.27, 0.03))$ in the punishment version of the two-stage task. *Top*: Distribution of the estimated parameter representing the degree of model-based control ($w$) (across 98 out of 100 simulations). *Bottom*: Mean ($\pm$SEM) proportion of stay choice at the first stage across 1000 simulations. **(B)** Mean ($\pm$SEM) proportion of stay choice at the first stage of the participants with high scores of OCI-R ($\geq 40$) in Experiment 1 (23 out of 548 participants) (top panel) and Experiment 2 (58 out of 1413 participants) (bottom panel) of [9], obtained by analysis of the data at https://osf.io/usdgt/. Cases where the "reward" at the previous trial was negative in the data file were omitted from the analysis.

panel). This pattern looks similar to the actually observed pattern based on 48 OCD subjects reported in the previous study [7] (their Supplementary Fig 1), although not similar to the pattern based on 32 OCD subjects reported in their preceding study [6] (their Supplementary Fig S1). Statistical test of this specific prediction is desired to be conducted in the future.

Looking at the choice pattern of OCD subjects in the punishment version of the two-stage task (Supplementary Fig 1 of [7]), apparently, OCD subjects tended to show smaller common-vs-rare differences in the "stay" proportion after no-punishment than after punishment. If punishment/no-punishment were equated with no-reward/reward considering the sign of RPEs, the direction of this asymmetry is opposite to the direction in the reward version of the task, and thus opposite to our model's prediction. However, if greater SR-based learning from positive dopamine signals suggested by our model occurs also in the circuit for RL of threat/ aversiveness, which is recruited in punishment contexts, as discussed above (Fig 8), such a direction is exactly predicted, as we confirmed by simulation of the aversive SR + appetitive IR agent (Fig 9A, bottom).

The data of the study examining the behavior in the two-stage task and self-report assessment of OCD using the Obsessive-Compulsive Inventory—Revised (OCI-R) [59] in general populations [9] is publicly available (https://osf.io/usdgt/). We extracted the data and analyzed the choice of participants with high scores of OCI-R ($\geq 40$) in their Experiments 1 and 2 (23 out of 548 and 58 out of 1413 participants, respectively). The choice in Experiment 2 (Fig 9B, bottom) showed a tendency of smaller common-vs-rare differences in the "stay" proportion after reward than after no-reward (comparison of $(P(\text{Stay})|_{\text{common}} - P(\text{Stay})|_{\text{rare}})|_{\text{reward}}$ and $(P(\text{Stay})|_{\text{rare}} - P(\text{Stay})|_{\text{common}})|_{\text{no-reward}}$, $t(57) = -1.99$, $p = 0.0516$ (Experiment 2); $t(22) = -1.31$, $p = 0.203$ (Experiment 1)). This direction of asymmetry is the same as the direction apparently

shown by OCD subjects in the punishment version, rather than the reward version [7]. We think this could potentially be explained as follows. In the study using general populations [9], "*Participants were paid a base rate (Experiment 1: $2, Experiment 2: $2.50) in addition to a bonus based on their earnings during the reinforcement-learning task (In each experiment, M = $0.54, SD = 0.04)* (sentence extracted from page 13 of [9])". Given that completion of the task (200 trials) would require a considerable amount of time, these payments were rather low. So it seems possible that the participants had expected to obtain much performance-dependent bonus and they felt every failure as a punishment, even though outcomes were facially presented as reward vs no-reward. Future study is desired to examine this possibility by manipulating the amount of payment. In terms of our model, the asymmetry in Experiment 2 can again be explained by the aversive SR + appetitive IR agent (Fig 9A, bottom).

## An alternative environmental model

The aversive SR + appetitive IR combination did not develop enhanced obsession-compulsion cycles in the environmental model that we used (proposed by [10]), as shown in the top-left corner of Figs 3E and 4H. If OCD patients have both appetitive SR + aversive IR and aversive SR + appetitive IR combinations in the parallel circuits for reward RL and threat RL, respectively, as proposed in Fig 8, obsession-compulsion of the patients could still potentially be explained by the former combination which developed enhanced obsession-compulsion cycles. However, a difficulty with this possibility is whether the circuit for reward RL, rather than that for threat RL, could be activated in such an aversive situation where the patients develop obsession-compulsion. Or even more generally, overgeneralization of (or longer memory trace for) positive, rather than negative, feedback that causes enhanced obsession-compulsion cycles in this environmental model might not be intuitively convincing, and can actually be inconsistent with (even opposite to) the previous work examining generalization in OCD patients [50] as discussed above.

A different possibility, then, is that the aversive SR + appetitive IR combination, in the threat RL circuit, could in fact develop obsession-compulsion if the environment is modeled differently. Indeed, in an alternative environmental model shown in Fig 10A, the aversive SR + appetitive IR combination developed intermittent bursts of repetitive obsession-compulsions, whereas the other combinations (appetitive SR + aversive IR, neutral SR + neutral IR, SR only, and IR only) rarely did so (Fig 10B and 10C). In this alternative environmental model, "compulsion" causes a stay at the obsession state with punishment whereas "depart" causes a transition to the relief state without cost or punishment, and so rational agents should learn to avoid "compulsion". Likewise, "intrusive" causes a transition to the obsession state with large punishment whereas "normal" causes a stay at the relief state without punishment, and so "intrusive" should normatively be minimized. However, the aversive SR + appetitive IR agent could overgeneralize the large punishing feedback upon "intrusive" (entering the obsession state) to preceding "depart" and "normal", deteriorating their values and increasing, in turn, the probabilities that "compulsion" and "intrusive" are chosen. Key differences from the original environmental model (Fig 2, [10]) are (i) punishment is given upon entering the obsession state, not only upon staying at it, (ii) "compulsion" could be repeated without returning to the relief state every time, and (iii) intrusive thought, rather than "abnormal reaction" to it, is modeled. Given these features, this alternative environmental model not only demonstrates that the aversive SR + appetitive IR combination could develop obsession-compulsion but also potentially overcomes several difficulties in the original environmental model discussed above.

This being said, however, consideration of such an alternative environmental model may be ad hoc. The agents' behavioral patterns in this model have parameter dependence.

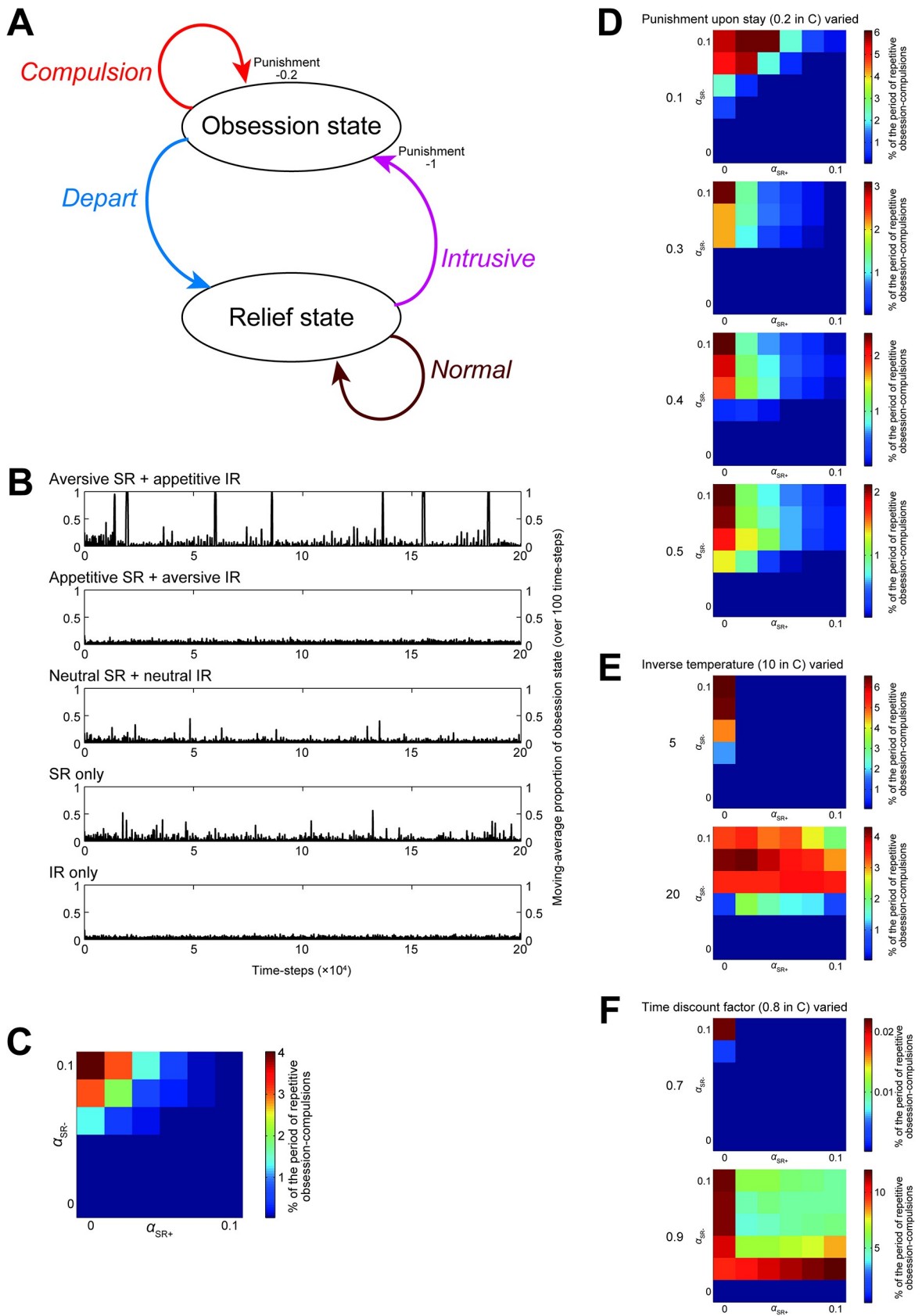

**Fig 10. Behavior of the dual-system agents in the alternative environmental model.** (A) Diagram of action-state transitions. At the relief state, the agent can have "normal (thought)" or "intrusive (thought)". Having "intrusive" causes transition to the obsession state, with punishment. At the obsession state, the agent can take "compulsion", which causes a stay at the obsession state with punishment, or "depart", which causes transition to the relief state. (B) Examples of the moving-average proportion of the obsession state (averaged over 100 time-steps, plotted every 100 time-steps) in the cases of the different types of agents (from top to bottom: aversive SR + appetitive IR ($\alpha_{SR+}, \alpha_{SR-}, \alpha_{IR+}, \alpha_{IR-}$) = (0.01, 0.09, 0.09, 0.01), appetitive SR + aversive IR (0.09, 0.01, 0.01, 0.09), neutral SR + neutral IR (0.05, 0.05, 0.05, 0.05), SR-only (0.1, 0.1, 0, 0) and IR-only (0, 0, 0.1, 0.1)). (C) Percentage of the period of repetitive obsession-compulsions, in which the moving-average proportion of the obsession state was $\geq 0.5$, during time-steps 0~50000 in various cases with different learning rates, averaged across 100 simulations for each case. The horizontal and vertical axes indicate $\alpha_{SR+}$ and $\alpha_{SR-}$, respectively, while $\alpha_{SR+} + \alpha_{IR+}$ and $\alpha_{SR-} + \alpha_{IR-}$ were kept constant at 0.1 (in the same manner as in Figs 3E and 4H). (D-F) Percentage of the period of repetitive obsession-compulsions in the cases where the size of punishment upon staying at the obsession state (originally 0.2 in (C)) was changed to 0.1, 0.3, 0.4, or 0.5 (D), the inverse temperature (originally 10 in (C)) was changed to 5 or 20 (E), or the time discount factor (originally 0.8 in (C)) was changed to 0.7 or 0.9 (F).

Specifically, while moderately changing the size of punishment upon staying at the obsession state (Fig 10D, top two panels) did not drastically change the patterns, further increasing the punishment upon stay (Fig 10D, bottom), changing the inverse temperature (Fig 10E), or changing the time discount factor (Fig 10F) caused significant changes: disappearance of (or prominent decrease in) the intermittent bursts of repetitive obsession-compulsions in the aversive SR + appetitive IR combination and/or appearance of them in other combinations. These results might rather indicate a fundamental limitation of modeling the environment for psychiatric disorders like OCD by such simple diagrams with only a few states and actions (c. f., [58]). Nonetheless, we would like to argue that the parallel opponent SR+IR configurations, which can be in line with the differential cortical targeting/activations of the direct and indirect basal-ganglia pathways and the existence of parallel cortico-basal ganglia-dopamine circuits for reward RL and threat RL, could potentially provide a novel biologically grounded framework to integrate the different lines of behavioral findings on OCD: memory trace imbalance, impaired model-based control, and its valence-dependence.

## Materials and methods

### Environmental model that describes possible enhancement of obsession-compulsion cycle

We adopted the environmental model developed by the previous study [10] that describes possible enhancement of obsession-compulsion cycle (Fig 2). There are two states, i.e., the relief state and the obsession state, and two available actions at each state, i.e., the "abnormal reaction" and the "other" action at the relief state and the "compulsion" and the "other" action at the obsession state. State transitions are determined as follows: at the relief state, taking the abnormal reaction always causes transition to the obsession state whereas taking the other action always causes stay at the relief state, and at the obsession state, taking the compulsion causes transition to the relief state and stay at the obsession state with 50% probability each whereas taking the other action causes transition to the relief state with 10% probability and stay at the obsession state with 90% probability. Taking the compulsion at the obsession state requires cost 0.01 (i.e., negative reward −0.01), and every stay at the obsession state imposes punishment 1 (i.e., negative reward −1); no other reward is assumed. The parameter values regarding state transitions and rewards are the same as those used in the previous work [10]. We also examined modified versions of the environmental model, in which there were two or eight "other" actions at both or one of the states (Fig 5). Probabilities of state transitions for these multiple "other" actions were set to the same values for the "other" actions in the original model. We further examined an alternative environmental model as depicted in Fig 10A.

## Agent having coupled SR- and IR-based systems

We adopted the model agent developed by the recent work [35] that has coupled SR- and IR-based systems (Fig 1C), with a modification that SR of actions (c.f., [13]) rather than SR of states was used in the present study because there were only two states in the environmental model. Each system developed the system-specific value of each action, and their average, referred to as the (integrated) action value, was used for action selection and TD-RPE calculation.

Action selection was made in a soft-max manner. Specifically, when there were multiple available actions $A_i$ ($i = 1,\ldots, n$), whose action values were $Q(A_i)$, $A_i$ was selected with probability

$$\exp(\beta Q(A_i))/\Sigma_i\{\exp(\beta Q(A_i))\},$$

where $\beta$ was the inverse temperature parameter representing the degree of choice exploitation over exploration and was set to 10 in the simulations with the abovementioned environmental models unless otherwise mentioned.

SARSA-type TD-RPE was used for the results presented in the figures. Specifically, when the agent took action $A_{t-1}$, paid cost $c$, received reward $r$, transitioned to the next state, and took the next action $A_t$, SARSA-type TD-RPE

$$\delta = -c + r + \gamma Q(A_t) - Q(A_{t-1}),$$

where $\gamma$ was the time discount factor and was set to 0.5 in the simulations with the abovementioned original environmental model (Fig 2) and its modified versions (Fig 5A) and 0.8 (or 0.7 or 0.9) in the simulations with the alternative environmental model (Fig 10A), was calculated. We also examined the cases with Q-learning-type TD-RPE for the original agent and environmental models, and the main properties of the agent's behavior shown in Fig 3 were not largely changed. Along with TD-RPE, TD errors for SR features were calculated:

$$\Delta = \mathbf{e}_{t-1} + \gamma \mathbf{M}_t, -\mathbf{M}_{t-1},$$

where $\mathbf{e}_{t-1}$ was the vector that had elements corresponding to all the actions and the element corresponding to $A_{t-1}$ was 1 and all the other elements were 0, and $\mathbf{M}_i$ ($i = t-1$ or $t$) was the $A_i$-corresponding row of the SR matrix $\mathbf{M}$.

IR-based system-specific value of $A_{t-1}$, $Q_{\text{IR}}(A_{t-1})$, was updated based on the TD-RPE:

$$Q_{\text{IR}}(A_{t-1}) \leftarrow Q_{\text{IR}}(A_{t-1}) + \alpha_{\text{IR}+}\,\delta \text{ (when } \delta \geq 0) \text{ or}$$

$$Q_{\text{IR}}(A_{t-1}) + \alpha_{\text{IR}-}\delta \text{ (when } \delta < 0),$$

where $\alpha_{\text{IR}+}$ and $\alpha_{\text{IR}-}$ were the learning rates in the IR-based system for positive (non-negative) and negative TD-RPEs, respectively. SR-based system-specific value of $A_{t-1}$, $Q_{\text{SR}}(A_{t-1})$, was given by the dot product of the SR feature vector for $A_{t-1}$, i.e., $\mathbf{M}_{t-1}$, and the weight vector $\mathbf{w}$, i.e.,

$$Q_{\text{SR}}(A_{t-1}) = \mathbf{w} \cdot \mathbf{M}_{t-1}.$$

The weight $\mathbf{w}$ was updated based on the TD-RPE:

$$\mathbf{w} \leftarrow \mathbf{w} + \alpha_{\text{SR}+}\delta\mathbf{M}_{t-1} \text{ (when } \delta \geq 0) \text{ or}$$

$$\mathbf{w} \leftarrow \mathbf{w} + \alpha_{\text{SR}-}\delta\mathbf{M}_{t-1} \text{ (when } \delta < 0),$$

where $\alpha_{SR+}$ and $\alpha_{SR-}$ were the learning rates in the SR-based system for positive (non-negative) and negative TD-RPEs, respectively. The SR feature vector for $A_{t-1}$, $\mathbf{M}_{t-1}$, was updated based on the TD errors for SR features:

$$\mathbf{M}_{t-1} \leftarrow \mathbf{M}_{t-1} + \alpha_{\text{feature}}\mathbf{\Delta},$$

where $\alpha_{\text{feature}}$ was the learning rate for feature update and was set to 0.01 in the simulations with the abovementioned environmental models except for the simulations shown in Fig 4D and 4E, in which $\alpha_{\text{feature}}$ decreased over time according to

$$0.01/(1 + \text{time-step}/1000).$$

For the simulations shown in Figs 4F–4H, 5 and 10, the IR-system-specific values for all the actions and all the elements of the weight vector for the SR system-specific values ($\mathbf{w}$) decayed at a constant rate (0.001), i.e., were multiplied by 0.999 at each time step.

The agent was initially placed at the relief state. The IR-system-specific values for all the actions were initialized to 0, and the weight vector for the SR system-specific values ($\mathbf{w}$) was initialized to $\mathbf{0}$. The SR matrix $\mathbf{M}$ was initialized to the identity matrix, i.e., the features corresponding to the own actions were initialized to 1 and all the other features were initialized to 0. 100 simulations were conducted for each condition. Below are notes on our simulations using the original environmental model. The value of the time discount factor ($\gamma = 0.5$) was the same as the one used in the previous work [10]. The ranges of the learning rates $\alpha_{IR+}$, $\alpha_{IR-}$, $\alpha_{SR+}$, and $\alpha_{SR-}$ were also determined in reference to the value of the learning rate used in that study (0.1), although because the average (rather than the sum) of the SR- and IR-based system-specific values was used for action selection and TD-RPE calculation, there was effectively a difference of two times, and also effective learning rates for the SR-based system were considered to vary as mentioned below. The value of the inverse temperature used in most simulations ($\beta = 10$) was larger than the value used in the previous work ($\beta = 1$) [10]. If $\beta$ was set to 1 in our original model without decay, the proportion that the agent was at the obsession state was rather high ($\geq$ around 0.5 on average) in any combinations examined in Fig 3A–3D (($\alpha_{SR+}$, $\alpha_{SR-}$, $\alpha_{IR+}$, $\alpha_{IR-}$) = (0, 0, 0.1, 0.1), (0.1, 0.1, 0, 0), (0.05, 0.05, 0.05, 0.05), and (0.09, 0.01, 0.01, 0.09)).

Notably, the abovementioned algorithm for TD learning of the SR-based system differed from the one described in [13], not only in the specific assumptions on the learning rates and the decay but also in that our algorithm lacked a scaling of the SR feature vector by its squared norm upon updating the weight. Without this scaling, the learning rate for weight update was effectively multiplied with the squared norm of the feature vector. In our simulations using the original environmental model, elements of the feature vectors were upper-bounded by $1 + \gamma + \gamma^2 + \ldots = 2$ (as we set $\gamma = 0.5$) and so the squared norm of the feature vector was upper-bounded by $2^2 = 4$, and since we varied the learning rates for weight update in the range of $\leq 0.1$, effective learning rates did not exceed 1.

In order to examine the reason for the value growth occurred in the original appetitive SR + aversive IR agent, we also examined the agent consisting of an appetitive IR-based system (($\alpha_{IR+}$, $\alpha_{IR-}$) = (0.09, 0.01)) and an aversive IR-based system (($\alpha_{IR+}$, $\alpha_{IR-}$) = (0.01, 0.09)). As in the case of SR + IR agent, the average of the two IR-based system-specific values, initialized to 0, was used for action selection and TD-RPE calculation.

## Simulation and fitting of the two-stage decision task

We simulated the behavior of the agent having a coupled SR- and IR-based systems in the two-stage decision task consisting of 201 trials [8]. There were two choice options at the first stage,

and selection of one of them led to either of two pairs of second-stage options with 70% or 30% probabilities (referred to as the common and rare transitions, respectively) whereas selection of the other first-stage option led to either of the two pairs with the opposite 30% and 70% probabilities. Selection of one of the second-stage options led to reward (size 1) or no-reward. The probability of reward for each second-stage option was determined according to Gaussian random walk with reflecting boundaries at 0.25 and 0.75, and it was implemented in a way that pseudo normal random numbers (mean 0, SD 0.025) was added to the probability at every trial and the probability was reflected at 0.25 and 0.75.

At every trial, choice was made in the soft-max manner with the degree of exploitation over exploration (inverse temperature) ($\beta$) set to 5 at both stages. SARSA-type TD-RPE for the first stage was calculated after the second-stage choice was determined:

$$\delta_k(1) = 0 + \gamma V(O_k(2)) - V(O_k(1)),$$

where $O_k(1)$ and $O_k(2)$ were chosen options for the first and second stages at the $k$-th trial, respectively, and $V(O)$ was the value of option $O$ (i.e., average of SR- and IR-system specific values). $\gamma$ was the time discount factor, and it was set to 1 (i.e., no temporal discounting). Then, after reward ($R = 1$ or $0$) was determined, TD-RPE for the second stage was calculated:

$$\delta_k(2) = R + 0 - V(O_k(2)),$$

where $V(O_k(2))$ reflected $\delta_k(1)$-based update of the weights of the SR system-specific values. The IR-based system-specific value of each first- and second-stage option was initialized to 0 and updated based on the TD-RPEs. The SR-based system had the SR of the first- and second-stage options, and the SR-based system-specific value was given by the dot product of the SR feature vector for the corresponding action and the weights, which were updated based on TD-RPEs. The SR matrix was initialized to the one that assumed the random policy on second-stage choice and incorporated the actual stage-transition probabilities:

$$\begin{bmatrix} 1 & 0 & 0.35 & 0.35 & 0.15 & 0.15 \\ 0 & 1 & 0.15 & 0.15 & 0.35 & 0.35 \\ 0 & 0 & 1 & 0 & 0 & 0 \\ 0 & 0 & 0 & 1 & 0 & 0 \\ 0 & 0 & 0 & 0 & 1 & 0 \\ 0 & 0 & 0 & 0 & 0 & 1 \end{bmatrix},$$

in which the first two rows/columns correspond to the two first-stage options and the subsequent four rows/columns correspond to the four (2×2) second-stage options. At every trial, after stage transition occurred and second-stage option was determined, SR features for the chosen first-stage option was updated based on the TD errors for SR features with the learning rate 0.05.

We considered an agent that had appetitive SR- and aversive IR-based systems, an agent that effectively had SR-based system only, and an agent that had neutral SR- and IR-based systems for which the learning rates in each system for positive and negative TD-RPEs were set to $(\alpha_{SR+}, \alpha_{SR-}, \alpha_{IR+}, \alpha_{IR-}) = (0.27, 0.03, 0.03, 0.27)$, $(0.3, 0.3, 0, 0)$, and $(0.15, 0.15, 0.15, 0.15)$, respectively, common to both stages, and conducted 100 simulations for each agent. We also conducted 100 simulations of the punishment version of the task, in which reward or no-reward ($R = 1$ or $0$) in the original task was replaced with no-punishment or punishment ($R = 0$ or $-1$), for an agent that had aversive SR- and appetitive IR-based systems (($\alpha_{SR+}, \alpha_{SR-}, \alpha_{IR+}, \alpha_{IR-}) = (0.03, 0.27, 0.27, 0.03)$).

Generated choice sequences were individually fitted by the RL model weighing model-based and model-free control described in the Supplemental material of the original study [8]. The RL model had seven free parameters: $a_1$ and $a_2$ (learning rates for the first and second stages), $b_1$ and $b_2$ (inverse temperatures for the first and second stages), $\lambda$ (eligibility trace, representing the degree with which the model-free value of first-stage option was updated based on the TD-RPE at the second stage), $\rho$ (perseveration or switching at first-stage choices), and $w$ (degree (weight) of model-based control, with the degree (weight) of model-free control set to $(1-w)$). Since $a_1$, $a_2$, $\lambda$, and $w$ should normally be between 0 and 1, these variables were transformed into $\mathrm{atanh}(2x-1)$, where $x$ represents $a_1$, $a_2$, $\lambda$, or $w$, in the fitting process so that the range of exploration was effectively bound to [0 1]. For each choice sequence, we explored a set of parameters that maximized the log likelihood [60] by using the "fminsearch" function of MATLAB, with the initial parameter values set as $(a_1, a_2, b_1, b_2, \lambda, \rho) = (0.3, 0.3, 5, 5, 0.2, 0.2)$ and $w = 0.2, 0.5$, and $0.8$ (i.e., these three cases were examined) and the 'MaxFunEvals' and 'MaxIter' options of this function set to both 10000. Among the three cases with different initial values of $w$, we selected a case that gave the maximum log likelihood, and if the "fminsearch" function terminated with "exitflag" 1 (meaning normal termination) in that selected case, the corresponding data (value of $w$) was included in the following analysis. This was true for 97, 97, 97, and 98 out of 100 simulations for the abovementioned four types of agents, respectively. We compared the obtained distribution of $w$ between the appetitive SR + aversive IR agent and the SR-only agent (originally planned), and also between the appetitive SR + aversive IR agent and the neutral SR + IR agent (additionally examined), by using the Wilcoxon rank-sum test (Mann Whitney U test).

We also simulated the behavior of the original RL model [8], as well as the behavior of a modified model with imbalanced eligibility trace, in the original reward version of the two-stage decision task. As for the original model, the parameters were set to $(a_1, a_2, b_1, b_2, \lambda, \rho, w) = (0.3, 0.3, 5, 5, 0.4, 0.2, 0.5)$. The modified model had two parameters, $\lambda_1$ and $\lambda_2$, representing the eligibility trace for positive (non-negative) and negative TD-RPEs, respectively, instead of the single parameter $\lambda$ in the original model, and they were set to $(\lambda_1, \lambda_2) = (0.4, 0)$ to represent imbalanced trace; the other parameters were set to the same values as those set for the original model. 100 simulations were conducted for each model, and generated choice sequences were individually fitted by the original model in the same manner as in the case of the agent with coupled SR- and IR-based systems described above. For each choice sequence, among the three cases of fitting with different initial values of $w$, we again selected a case that gave the maximum log likelihood, and this time the "fminsearch" function terminated with "exitflag" 1 in that selected case for all of the 100 simulations for both models. We again compared the obtained distribution of $w$ for each model by using the Wilcoxon rank-sum test.

Precisely speaking, our implementation of the RL model developed in [8] and its variant, used for the fitting and simulations mentioned above, could be slightly different from the original one. Specifically, we could not find how the model-based system estimated the transition probabilities when the actual frequencies of past occurrence of both types of transitions exactly matched (for example, both types of transitions occurred 10 trials in the initial 20 trials) in the original study, and we assumed that the model-based system estimated them as 50% and 50% in such situations.

In addition to the analyses mentioned above, we separately conducted 1000 simulations for each of the six cases mentioned above, and analyzed the choice pattern, i.e., calculated the proportion of stay choice (i.e., choosing the same option as the one chosen at the previous trial) at the first stage depending on whether reward, or punishment, was obtained or not and whether common or rare transition was occurred at the previous trial (we increased the number of simulations to 1000 in order to obtain the average pattern more accurately).

## Simulation and fitting of the delayed feedback task

We simulated execution of the delayed feedback task [10] by the dual-system agents. There were eight stimuli, which caused +40, +10, −10, or −40 immediate feedback (denoted as +40$^{im}$ etc) or +40, +10, −10, or −40 delayed feedback three trials later (denoted as +40$^{de}$ etc). The task consisted of six sessions, and in each session, each of six pairs of stimuli ((+10$^{im}$, +40$^{im}$), (+10$^{de}$, +40$^{de}$), (−10$^{im}$, −40$^{im}$), (−10$^{de}$, −40$^{de}$), (+10$^{im}$, +40$^{de}$), (−10$^{im}$, −40$^{de}$)) was presented ten times and each of ten pairs ((+40$^{im}$, +40$^{de}$), (+10$^{im}$, +10$^{de}$), (−10$^{im}$, −10$^{de}$), (−40$^{im}$, −40$^{de}$), (+10$^{de}$, +40$^{im}$), (−10$^{de}$, −40$^{im}$), (+10$^{im}$, −10$^{im}$), (+40$^{im}$, −40$^{im}$), (+10$^{de}$, −10$^{de}$), (+40$^{de}$, −40$^{de}$)) was presented five times in a pseudorandom order (in the same manner as in the original study [10]).

The IR-based system of the dual-system agent learned the system-specific values of eight actions to choose each of the eight stimuli through updating the value of taken action by RPE multiplied by the learning rate ($\alpha_{IR+}$ or $\alpha_{IR-}$ for positive or negative RPE, respectively). The SR-based system represented each action by estimated discounted cumulative occupancies of successor "states". Seven states were defined based on the presented (total) feedback at each trial: $S_1$: +80, $S_2$: +50 or +40 or +30, $S_3$: +20 or +10, $S_4$: 0, $S_5$: −10 or −20, $S_6$: −30 or −40 or −50, and $S_7$: −80. At each trial $k$, "trace" of each action $A_i$ ($i = 1,.., 8$), $T(A_i)$, was updated as

$$T(A_i) \leftarrow \gamma T(A_i) \text{ (if } A_i \text{ was not taken at trial } k) \text{ or}$$

$$T(A_i) \leftarrow \gamma T(A_i) + 1 \text{ (if } A_i \text{ was taken at trial } k),$$

where $\gamma$ was the time discount factor set to 0.75. Then, a column of 8×7 SR matrix **M** that corresponded to the state at trial $k$ (determined by the feedback presented at trial $k$), $S_j$, was updated to be equal to the traces for each action:

$$\mathbf{M}_{ij} \leftarrow T(A_i) \ (i = 1,.., 8),$$

after **M** was used for value calculation and value (weight) update (mentioned below). The other columns of **M** remained unchanged. The SR-based system-specific value of $A_i$ ($i = 1,.., 8$) was given by the dot product of the SR feature vector for $A_i$, i.e., $i$-th row of **M**, $\mathbf{M}_{(i)}$, and a weight vector **w**, i.e., $\mathbf{w} \cdot \mathbf{M}_{(i)}$. The weight **w** was updated based on the action $A_i$ that was taken at trial $k$ and RPE $\delta$:

$$\mathbf{w} \leftarrow \mathbf{w} + \alpha_{SR+}\delta\mathbf{M}_{(i)} \text{ (when } \delta \geq 0) \text{ or}$$

$$\mathbf{w} \leftarrow \mathbf{w} + \alpha_{SR-}\delta\mathbf{M}_{(i)} \text{ (when } \delta < 0).$$

Integrated action values were calculated as the average of the IR- and SR-based system-specific action values. Choice between two actions corresponding to two stimuli presented at each trial was made in the soft-max manner with the degree of exploitation over exploration (inverse temperature) ($\beta$) set to 0.25. RPE was calculated as the difference between the (total) feedback and the integrated value of taken action. The IR-based system-specific values for all the actions and all the elements of the weight vector for the SR-based system-specific values (**w**) decayed at a constant rate (0.001), i.e., were multiplied by 0.999 at each trial. The traces for actions $T(A_i)$, the elements of the SR matrix **M**, and the elements of the weight vector **w** were all initialized to 0. The learning rates were set as ($\alpha_{SR+}$, $\alpha_{SR-}$, $\alpha_{IR+}$, $\alpha_{IR-}$) = (0.02, 0.02, 0, 0) or (0.018, 0.002, 0.002, 0.018) for the SR-only agent or the appetitive SR + aversive IR agent, respectively. We conducted 1000 simulations for each of the two types of agents, and calculated the averages and the SEMs.

We also conducted fitting of the choices of these two types of dual-system agents (50 simulations for each, separately conducted from the abovementioned 1000 simulations) by the separate eligibility trace actor-critic model considered in [10]. This model had four free parameters: $a$ (learning rate), $b$ (inverse temperature for choice), and $v^+$ and $v^-$ (decay time scale of the trace for positive and negative RPEs, respectively). Since $a$, $v^+$, and $v^-$ should normally be between 0 and 1, these variables were transformed into atanh($2x-1$), where $x$ represents $a$, $v^+$, or $v^-$, in the fitting process so that the range of exploration was effectively bound to [0 1]. For each choice sequence, we explored a set of parameters that maximized the log likelihood [60] by using the "fminsearch" function of MATLAB, with the initial parameter values set as ($a$, $b$) = (0.0011, 2.96), which were the medians of estimated parameters of HCs in [10], and $v^+$, $v^-$ = 0.1, 0.5, or 0.9 (i.e., $3 \times 3 = 9$ cases were examined) and the 'MaxFunEvals' and 'MaxIter' options of this function set to both 10000. Among the nine cases with different initial values of ($v^+$, $v^-$), we selected the cases in which the "fminsearch" function terminated with "exitflag" 1 (meaning normal termination), and if there were any such case(s) (this was true for 47 and 45 simulations for the SR-only and the appetitive SR + aversive IR agent, respectively), we selected a case that gave the maximum log likelihood among them, and plotted the corresponding values of $v^+$ and $v^-$ in Fig 7C.

## Software and statistics

Simulations and fitting were conducted by using MATLAB, and its functions "rand", "randn", and "randperm" were used to implement probabilities and pseudorandom numbers. Standard deviation (SD) of simulation data was normalized by $n$, and standard error of the mean (SEM) was approximately calculated as SD/$\sqrt{n}$. The Wilcoxon rank-sum test (Mann Whitney U test), Shapiro-Wilk normality test, and paired t test were conducted by using R (functions "wilcox. test", "shapiro.test", and "t.test").

## Code availability

Codes to generate/reproduce the data presented in the figures are available at: https://github. com/kenjimoritagithub/sr101.

## Supporting information

**S1 Data. Codes to generate/reproduce the data presented in the figures.** Please read "readme.txt" for details.
(ZIP)

## Author Contributions

**Conceptualization:** Kenji Morita.

**Funding acquisition:** Kenji Morita.

**Investigation:** Reo Sato, Kanji Shimomura, Kenji Morita.

**Writing – original draft:** Kenji Morita.

**Writing – review & editing:** Reo Sato, Kanji Shimomura, Kenji Morita.

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
