## [Decision Letter · Decision Letter 0]

26 Feb 2023

Dear Dr Morita, 

Thank you very much for submitting your manuscript "Opponent Learning with Different Representations in the Cortico-Basal Ganglia Pathways Can Develop Obsession-Compulsion Cycle" for consideration at PLOS Computational Biology.

As with all papers reviewed by the journal, your manuscript was reviewed by members of the editorial board and by several independent reviewers. In light of the reviews (below this email), we would like to invite the resubmission of a significantly-revised version that takes into account the reviewers' comments.

As you will see the reviewers raised many major points. We will ask to carefully take into account while revising the manuscript. I also encourage to take this guidelines into account while preparing the rebuttal letter: https://psyarxiv.com/kyfus

We cannot make any decision about publication until we have seen the revised manuscript and your response to the reviewers' comments. Your revised manuscript is also likely to be sent to reviewers for further evaluation.

Sincerely,

Stefano Palminteri

Academic Editor

PLOS Computational Biology

Daniele Marinazzo

Section Editor

PLOS Computational Biology

As you will see the reviewers raised many major points. We will ask to carefully take into account while revising the manuscript. We will also suggest that in the revised manuscript to not separate figures from the main text and captions from figures. It makes the paper much more challenging to read through. I also encourage to take this guidelines into account while preparing the rebuttal letter: https://psyarxiv.com/kyfus

Reviewer's Responses to Questions

**Comments to the Authors:**

Reviewer #1: Here, the authors attempt to reconcile two current theories underlying obsessive-compulsion - the impairment of the model-based system with a shorter memory trace for negative (than positive) PE. They simulated a model with an agent containing an SR (model-based) and IR (model-free) system, and adjusted its learning based on negative and positive PE at different rates. They found that if the agent's SR system learned primarily from positive PEs, while the IR system learned primarily from negative PEs, the agent would result in an increased looping of the obsession-compulsion states. Further, they simulated the behavior of this (SR+IR) agent completing the two-step task, vs an agent that only has the SR system. They found that the SR+IR agent had lower model-based control than the SR agent. Overall, the authors suggest that the opponent learning of positive PEs for SR and negative PEs for IR may underlie obsession-compulsion.

I thank the authors for an interesting manuscript. I thought the work was quite intriguing, and is indeed a step towards delineating mechanisms underlying OCD, extending both the goal-directed theory and the memory trace theory. I also appreciate the many details that went into the work. I do think the manuscript could work with some tweaks to the grammar and overall rewriting to increase the readability. The results could work with better signposting on the motivation of the analysis - especially since the methods are at the end, it requires more context. Finally, there are a lot of dense concepts which are dropped with minimal explanation and referred to back and forth - where further elaboration/explanation would help to hone in on the key messages put forward. More specific comments are as follows:

Major points

1) The introduction starts off very curt, stating the two theories under investigation of the paper, but does not really explain why the theories are related to OCD. How is impaired goal-directed control related to OCD symptoms? Perhaps some reference to dual system theories might help to set the scene. How does having a short memory trace for negative PEs also contribute to OCD? How does it affect learning and therefore behavior (e.g., lower punishment learning, therefore make more wrong actions)?

2) It would be more cohesive if SR and IR were introduced in parallel to model-based/model-free - currently there is a link between SR and model-based control, but IR link to model-free only comes into the last paragraph

3) "Moreover, we also expected that fitting of behavior of the agent with combined appetitive SR- and aversive IR-based systems could potentially result in smaller weights of model-based control than the agent with SR-based system only in the two-stage decision task" - isn't this automatically expected because SR, as the point was made above, is a model-based system, and thus would have greater model-based control vs a SR-IR (includes model-free) system?

4) It would help if Figure 1A included the outcomes, to contrast that the IR has a direct representation of A1 to final outcome.

5) "whereas eligibility-trace imbalance per se would not bias the estimation of the degree of model-based control." - so is the point that it requires both the model-based and model-free system, in combination with a eligibility-trace imbalance, to underlie OCD symptoms (vs just an imbalanced trace in the model-based system)? This seems to a main finding, and is not very well motivated in the introduction, so the analysis/results come as a surprise.

6) The description of the environmental model was fairly clear, although in its figure, the loop in the obsession state to itself is confusing because the "other" label is very close to it

7) Behavior of the dual-system agent in the environmental model - authors refer to Figure 3 indicating what the figure shows generally, but not its trends i.e. "Figure 3A-top shows an example of the time evolution of the difference in the (integrated) action values of "abnormal reaction" and "other" at the relief state (Qabnormal − Qother@relief)." but instead could described as "proportion of obsession state stays low/below threshold, indicating that the agent minimized selection...etc, etc". I think it would help in actually describing the results in the text, because with the current descriptions it does not tell me anything except that I have to refer to the figures. Also the bottom Figure 3 y-axes should indicate it is the average of the middle one- it is a bit misleading to have them the same y axis label.

8) Similar comments as above to "Figure 3D show the results."

9) In Figure 6, y-axes are missing, and it would nice to add a distribution curve since the authors are comparing distributions

10) "As shown in the left panels of Figure 6A,B, the distribution of the estimated w for the agent with appetitive SR- and aversive IR-based systems (Figure 6B) was smaller than that for the agent with SR-based system only (Figure 6A)." Consider rephrasing "smaller distribution"; does this refer to the variance/mean peak? Looks like Figure 6B ranges from 0 to 1, similar to Figure 6A.

11) "In contrast, fitting of choices generated by the original RL model with balanced eligibility trace (λ = 0.4 for both positive and negative RPEs) and those generated by a modified model with imbalance trace (λ = 0.4 and 0 for positive and negative RPEs, respectively), both of which had the same degree of model-based control (w = 0.5), did not result in smaller estimated w in the latter, as shown in the left panels of Figure 6C,D" - This analysis needs to be more clearly motivated. What is the point of testing the eligibility trace difference after the SR-IR system one, and what is expected from this? Also a bit unclear on if both models were specified with the same degree of w, why would it possibly result in differing w?

12) I wonder if it would be a fairer comparison to contrast the appetitive SR and aversive IR model with other versions of the SR-IR model (e.g., differing with pos/neg PE learning) like the models in the paragraph above. This could highlight the contribution of the appetitive/aversive component of the SR-IR model, which presumably is the main aspect that underlies the OC cycle.

13) "We further analyzed the choice pattern of each type of agent. The agent with appetitive SR and aversive IR-based systems made a model-based-type choice, taking into account whether the occurred state transition was the common type or rare type, after reward was obtained but much less so after reward was not obtained (Figure 6B, right panel). In contrast, the SR-only agent, as well as the agents with balanced or imbalanced eligibility trace, made a model-based-type choice regardless of whether reward was obtained at the previous trial (Figure 6A,C,D, right panels)." This paragraph requires rephrasing and actual explanation of what the pattern of choices mean. Model-based behavior dominates in Figures 6A, C and D vs the influence of model-free in Figure 6B- it is inaccurate to call it "making model based choice", because it is not about a single choice but stay probability over all trials.

14) Overall, the study presents some convincing evidence to the appetitive SR and aversive IR role underlying the OC cycle, particularly the demonstration of how the loop gets stuck in the obsessional-compulsion states. However I think there could be more comparisons with the models applied to the two step task, the difference between and SR system only and the appetitive/aversive IR system seem to have many components that would contribute to the difference in behavior, and it would be clearer to delineate them.

15) There is some nuance in the expression of OCD that is not fully taken into account of the current model. OCD patients present varying levels of obsessions vs compulsions, e.g. pure obsessional/compulsive OCD. Whereas the current model assumes that it is a cycle of the obsessive state to the compulsion state (to resolve distress), where both states are necessary. I wonder how these other presentations could be accounted for. Another point to raise is that there is a hypothesis specifically borne from the goal-directed work of compulsions (Gillan & Sahakian, Neuropsychopharmacology, 2015), it is suggested that obsessions in OCD might arise as a result of compulsive behavior instead - what do the authors think of this in terms of their current work?

16) "Specifically, we could not find how the model-based system estimated the transition probabilities when the frequencies of past occurrence of both types of transitions exactly matched in the original study, and we assumed that the model-based system estimated them as 50% and 50% in such situations" - Not quite sure what the authors mean here - common vs rare transition is 70/30%, and so the occurrence of the transitions should match that as well? In terms of simulations, the transition matrix would start with 50% for all types of transitions and updates trial by trial based on the choices made in the task.

Minor points

17) The authors refer to 'agent' without an article (an/the) throughout the text, it reads a bit weird without one

18) "Specifically, modeling showed that obsession-compulsion cycle can be enhanced in agent having much shorter memory (eligibility) trace for negative than positive prediction errors (PEs), and fitting of behavioral choices suggested that OCD patients indeed had such an imbalance as compared to healthy controls (HCs), although even HCs tended to show shorter memory trace for negative than positive PEs." This sentence is way too long and is very confusing. Consider splitting it into more sentences.

19) "...this possibility explains why human is prone to OCD and even healthy people tended to show shorter memory trace for negative PEs." - should be 'humans are'. More broadly I think this these two conclusions can be phrased better.

20) "and fitted them by the RL model weighing model-based and model-free controls developed in the original study" - it should just be 'control' (controls can be misinterpreted as healthy controls)

Reviewer #2: The manuscript describes an interesting integrative model of OCD, that has the potential to explain both clinical phenomena (persistent obsessions) and well-known two-stage task findings, based on the idea that patients’ model-based system (or more specifically - SR) learns disproportionately more from positive RPEs, whereas their model-free system learns more from negative RPEs. I applaud the authors for the attempt to integrate previous computational models of OCD – the field is in desperate need of such endeavors. Furthermore, the model is sophisticated and appears to be biologically plausible.

That being said, I believe that several major conceptual, empirical, and style-related issues limit the potential contribution and impact of the manuscript.

First, the model is justified chiefly based on some overlapping neurobiological findings concerning different distinctions between direct and indirect pathways of the basal ganglia. Although this seems reasonable, I believe the manuscript/model lacks sufficient intuitive psychological/clinical justification. What is the psychological interpretation of the model? Is it the idea that OCD patients are more likely to start obsessing because they essentially overweight the probability that they will be able to ‘exit’ the obsessing cycle via compulsions (such that the SR/MB value of entering it is not as negative as it could have been)? Whether this is the core idea, or a different explanation would be more precise– I believe such an explanation must be repeatedly spelled out in the introduction, and the interpretation of the results. Insufficient interpretability is also evident, for example, regarding the finding that the alternative model with 8 ‘other’ actions does not lead to obsessive phenomenology (also, can this result be attributed specifically to the existence of many ‘other’ actions starting form the relief state or, alternatively, the ‘other’ actions starting from the obsession state?).

In general, such interpretability is important for two main reasons. First, although this is a computational paper and a computational journal, I believe its conceptual assumptions/conclusions must be interpretable by OCD experts lacking the extensive technical background the current conceptualization requires. Second, spelling out the conceptual assumptions/conclusions will enhance the ability to evaluate the psychological plausibility of the model.

This raises a second issue that focuses on the specificity of the results to the specific environmental model depicted in Figure 2. A fundamental assumption (or maybe a result) of the model is that the mechanism of obsessive-compulsive behavior is related to a higher probability of having an ‘abnormal reaction’, or, in other words – a higher probability of experiencing obsessions. This is interesting but also not somewhat controversial. Some classical models of OCD will argue that obsessions (or intrusive thoughts) occur universally and that the core problem in OCD is either a higher probability of staying in the obsessional state or alternatively, a higher probability of using compulsions to exit it. Is there any evidence for such ‘behaviors’ in the model? For example, does the `OCD agent` use compulsions as a main policy (vs. ‘other’) to exit the obsessional state? If we examine the two agents (with/without OCD) only in cases/steps in which they have entered the obsessional state – is the OCD agent more likely to use compulsions to exit it? Furthermore, is the ‘OCD agent’ more prone to ‘get stuck’ in the obsessions state, or is the general increase in obsession caused by intermittent obsessions-compulsion cycles? More generally - how do excessive learning from positive SR RPEs and reduced learning from positive IR RPEs affect policies starting from the obsession state?

Third, an interesting result of the simulations is that the probability of experiencing an obsession, even in an `agent with OCD` never surpasses 50%. This raises two questions. First – how plausible is this result? Oftentimes, OCD symptoms can become so dominant in a specific context that patients will tend to experience obsessions and emit a compulsion in close to 100% of the times they find themselves in this context (e.g., while handwashing, or when leaving the house, etc.). Second – what does this entail regarding the theoretical role of the temperature parameter in the model? Isn’t it the case that a nearly-deterministic model here will never move to the obsessional state?

Fourth, I believe the ability of the model to provide an alternative explanation for the common finding of reduced model-based learning in the two-step task is intriguing and important. I also applaud the researchers for discussing the limitations of this alternative interpretation vis-à-vis other findings in this literature. I have two questions here. First, can the model also explain findings concerning model-based learning in simple structure learning tasks (e.g., Sharp, Dolan & Eldar, 2021, Psychological Medicine)? Second, it would help the paper dramatically if the authors were able to fit their model to actual two-step task data (some are publicly available, I believe, e.g., https://osf.io/usdgt/)

Finally, I have to say that I found the description of the models hard to follow. The methods section could benefit from a more detailed, step-by-step exposition of the models. From a purely stylistic point-of-view – it would be much easier to read and understand the equations if they were presented on separate lines, where all components are explained.

A few minor comments:

• The introduction could benefit if the relatively new ‘trace imbalance model’ were explained in more detail. I realize that it was developed and validated in a separate paper, but IMO, the readers should be able to understand it without having to read that other paper.

• Figure 2 could benefit if the relevant parameters controlling the probability for abnormal reaction/compulsion are explicitly stated (and thus linked to the equations)

• Some intuitive explanation regarding the ‘weights’ of the SR system mentioned in the results (and figures) can be helpful.

• On page 8, the authors say, “Now, assume that the agent, developing enhanced obsession-compulsion cycle, exists at the relief state and happens to take the "other" action repeatedly, without taking the abnormal reaction. It is in fact the optimal policy, and because no reward or punishment is given as long as the agent only takes the "other" action, the value of the "other" action should approach, through RPE-based update, to 0, which is equal to the optimal value. However, this value, 0, is lower than the positive value of the abnormal reaction (even though it slowly decays)”. This is an interesting result. My main question is whether we could also predict the opposite in the case of an agent of chooses the abnormal reaction repeatedly?

• The idea that the learning rate biases in the SR/IR systems are ‘evolutionarily selected’ is interesting, but I couldn’t understand the exact proposed function of this setting. The authors cite a paper examining this in dynamic environments, but it would be helpful if they could briefly explain the logic here.

• In my opinion, the limitations section, and especially the second limitation – is too specific. I believe this section would benefit if it would focus on more conceptual-level limitations (e.g., what clinical/psychological phenomena the model cannot explain? To what extent are the simulations results dependent upon specific fixed parameters or settings of the model's environment?

Reviewer #3: The paper by Sato and colleagues proposes a computational reinforcement learning model of obsessive compulsive disorder symptoms. The model proposes a reinforcement learning agent that learns to predict action values from multiple representations: one over successor representations (SR) and one over independent representations (IR). The key feature of the model which the paper proposes can underlie obsessive compulsive symptoms is that learners for the SR and IR representations differentially learn from positive and negative reward prediction errors. Specifically, the SR learns more from positive reward prediction errors, and the IR learns more from negative reward prediction errors.

The claim of the paper is that this model can link together two separate findings relating to decision-making and OCD symptoms. The first is findings which that individuals with OCD have higher eligibility traces for positive compared to negative learning. The authors account for these findings by showing that, like a model with imbalanced eligibility traces, their model can also generate obsessive-compulsive cycles in a toy two-state MDP example.

The second is that individuals with OCD show less model-based learning on the two-step task. This is accounted for by showing that their model demonstrates less model-based behavior than an agent that decides just using the SR.

Overall, I think the attempt to relate compulsive symptoms to the types of representations that individuals employ in reinforcement learning is commendable and also could be fruitful approach. However, I have a number of critiques and suggestions for further work for the specific simulations presented in the paper.

1. Simulation of Sakai et al., 2022. The first part of the paper attempts to offer a re-interpretation of previous findings from Sakai et al., 2022, which argued that individuals with OCD have a higher eligibility trace for positive compared to negative prediction errors. The current paper argues that a model where an SR learner has a higher learning rate for positive prediction errors compared to negative prediction errors, and an IR learner has the opposite pattern, can generate equivalent behavior. This is demonstrated by simulating behavior in a toy MDP and showing the model can develop obsessive-compulsive cycles that were generated in Sakai et al., 2022.

I think to make the argument though that an SR + IR learner can account for the model of Sakai et al., 2022, it is also necessary to demonstrate that the new model can generate the behavior of OCD individuals (and variation between OCD and healthy controls) from the empirical study presented in Sakai et al, 2022 which actually provides evidence for the eligibility trace account. Can an SR an IR based learner with different learning rates for positive versus negative prediction errors explain how controls varied from individuals with OCD in the experiments that were run in Sakai et al., 2022? This should be demonstrated with simulation if the authors want to claim that their new model can capture what the previous model was suggested to capture.

2. Updates for SR system. I think something is going a bit wrong with how the SR learning is implemented, which is leading it to somewhat bizarre behavior in the simulation of obsession-compulsion cycle. This is clear from the weights and value estimates that are being learned by the SR, and plotted in Fig. 4B and D. In particular, it should not be possible for the value of any action to be above 0, since there are only negative rewards in the task.

My guess is that the cause of such divergent large weights is in how weights for the SR-based system are being updated, and that the representation is not being appropriately scaled for the update. This in turn is causing learning rates that are effectively above 1. Positive prediction errors with learning rates greater than 1 can cause the SR learner to form positive value estimates even when there are no positive rewards in the task.

To explain this more technically: the equation that is used to learn weights for the SR system is w_new <- w_old + M_transpose*delta*alpha, where M is the feature vector (a row vector here) for the state from which the action was taken, delta is the reward prediction error, and alpha is the learning rate. Typically, when implementing this update, it is necessary to scale the feature vector for the update. Specifically, M needs to be divided by its dot-product with itself (M*M_transpose). This is reported in the methods section of Russek et al., 2017, PLOS Computatainal Biology, which uses this update to simulate some aspects of MB behavior. Without scaling the feature vector, the learning rate basically loses meaning, and it can wind up effectively being above one.

To explain this, consider the usual TD update, for tabular TD:

V_new <- V_old + alpha*delta,

where alpha is the learning rate and delta is the prediction error. Here, alpha has an interpretation as the amount that V changes as a proportion of the prediction error:

(V_new - V_old) = alpha*delta.

For the linear-feature function approximation, applying the same idea, we have

(V_new - V_old) = M*w_new - M*w_old.

For the update, w_new <- w_old + M*delta*alpha, if M is not scaled appropriately, the net change to V after the update is:

(V_new - V_old) = M*w_new - M*w_old

= M(w_new - w_old) -- factoring out M

= M(w_old + M_transpose*delta*alpha - w_old) -- replacing w_new with the update rule

= M*M_transpose*delta*alpha -- simplifying and multiplying in M)

So, effectively, the change in V is no longer determined by the learning rate alpha, but rather by M*M_transpose*alpha. Thus, for alpha to have the same meaning it has in tabular TD, the feature vector thus needs to be scaled in the update by M*M_transpose.

Without this scaling, when components of M are above 1 (which is what happens in tasks where states can be visited multiple times, as in the OCD example) the effective learning rate can take on values greater than 1. This can cause the SR-system’s values to go above 1 in tasks where all rewards are negative. This also, I think, is what causes the weights to not converge, but rather keep growing over time.

It is possible that doing these sorts of non-scaled divergent updates might be a hypothesis for some aspects of compulsive behavior, but I think it is not the one that the authors intend to test in the paper. I’d think they should thus try to re-do the simulations with scaled updates.

Alternatively, if this scaling is not the reason for the SR weights to grow so divergently, I think it is important for the authors to discover why the SR behaves so erratically in this task. This is necessary to understand how the relevant obsessive-compulsive behavior is being produced.

3. Two-step task simulation. For the two-step task simulation, the authors compare their proposed model of compulsion (SR + IR with asymmetric updating) to a model which only uses the SR. They find that SR + IR with asymmetric updating produces less model-based behavior than SR alone.

This finding however is not surprising, and does not provide evidence that the proposed model (SR + IR with asymmetric updating) produces OCD behavioral patterns in the two-step task. Applied to the two-step task, the SR functions like a model-based learner and the IR functions like a model-free learner. The simulations thus effectively compare a system which combines MB (here SR) and MF (here IR) to a system that only uses MF (IR). The simulations thus don’t test the need for asymmetric updating in producing OCD-like behavior (less MB). To make the argument that the proposed model generates observed OCD behavior, the authors need to compare SR and IR with asymmetric updating to SR and IR with neutral updating, and see whether this produces less MB behavior, in a manner consistent with findings on behavior from OCD individuals. This is needed to validate the claim that asymmetric updating to SR vs IR systems generates compulsive symptoms.

4. Possible additional empirical analysis. This is a suggestion that the authors do not need to complete to make a publishable paper, but I think would make the paper much stronger, so they might consider. The two-step task simulations have clear predictions of what compulsive behavior in the two-step task should look like (more MB behavior following rewards compared to non-reward). This prediction could be tested in existing public datasets of the two-step task. Specifically, data from Gillan et al., 2016, which analyzed how two-step behavior varies with self-reported compulsivity symptoms is publicly available at https://osf.io/usdgt/. It should be possible to re-analyze this data and determine whether OCD symptoms are associated with more asymmetric updating of an IR + SR learner (or alternatively whether there is more MB behavior following reward compared to non-reward events in individuals higher in self-reported OCD symptoms). Such an empirical result would make the argument in the paper much stronger.

**Have the authors made all data and (if applicable) computational code underlying the findings in their manuscript fully available?**

Reviewer #1: Yes

Reviewer #2: **No: **I wasn't able to find the supplementary material (which should include code), so I answered 'no' to the question regarding code/data availability below. But perhaps this is just a technical issue in the system.

Reviewer #3: None

PLOS authors have the option to publish the peer review history of their article (what does this mean?). If published, this will include your full peer review and any attached files.

Reviewer #1: No

Reviewer #2: No

Reviewer #3: No
---

## [Decision Letter · Decision Letter 1]

8 May 2023

Dear Dr Morita, 

Thank you very much for submitting your manuscript "Opponent Learning with Different Representations in the Cortico-Basal Ganglia Pathways Can Develop Obsession-Compulsion Cycle" for consideration at PLOS Computational Biology.

As with all papers reviewed by the journal, your manuscript was reviewed by members of the editorial board and by several independent reviewers. In light of the reviews (below this email), we would like to invite the resubmission of a significantly-revised version that takes into account the reviewers' comments.

While all the reviewers appreciated your efforts and clarifications, two reviewers (1 and 3) still present significant and reasonable doubts concerning the potential impact and solidity of your arguments. So, unfortunately, at this stage we cannot accept and the paper and we have to ask to further revise the paper while taking into account the remaining points of the reviewers (especially 1 and 3). We apologize for the quite long delay and the additional load of work that it will require, but in line with PLoS Computational Biology standard and in order to make sure that the paper delivers its promises. We specifically stress the fact that

1) to which extent the features of the models contribute to the model behavior and specific pattern (OCD cycles)

2) the stability of the results as a function of the choice of the parameters

We cannot make any decision about publication until we have seen the revised manuscript and your response to the reviewers' comments. Your revised manuscript is also likely to be sent to reviewers for further evaluation.

Sincerely,

Stefano Palminteri

Academic Editor

PLOS Computational Biology

Daniele Marinazzo

Section Editor

PLOS Computational Biology

While all the reviewers appreciated your efforts and clarifications, two reviewers (1 and 3) still present significant and reasonable doubts concerning the potential impact and solidity of your arguments. So, unfortunately, at this stage we cannot accept and the paper and we have to ask to further revise the paper while taking into account the remaining points of the reviewers (especially 1 and 3). We apologize for the quite long delay and the additional load of work that it will require, but in line with PLoS Computational Biology standard and in order to make sure that the paper delivers its promises. We specifically stress the fact that

1) to which extent the features of the models contribute to the model behavior and specific pattern (OCD cycles)

2) the stability of the results as a function of the choice of the parameters

Reviewer's Responses to Questions

**Comments to the Authors:**

Reviewer #1: I thank the authors for their comprehensive revision - the introduction set the context and background with clear motivations and hypotheses, and the results were also signposted and explained well. I have no further questions on the points that were raised previously, only minor question/suggestions:

1. The addition of the inverse temperature analysis is quite interesting. I think there should be a little further elaboration of the role/effect of the temperature parameter in this context - does this mean individual differences in exploration/exploitation could predict whether the agent (and presumably, a human) would go into the obsessional state/develop OCD?

2. In the analysis for the choice patterns of the agents (Figure 6), I wonder if it would be helpful to report some statistics either with the P(Stay) difference or the reward*trans effect from the LMM (i.e., Stay ~ Reward * Transition + (Reward * Transition + 1 | Subject)), as the stay probability graphs differences are quite small.

Reviewer #2: I appreciate the authors' considerable efforts in revising the manuscript, particularly the additional analyses examining the questions the other reviewers and I raised. Indeed, the additional analyses and revisions clarify and improve the manuscript.

However, these important clarifications and analyses also emphasize some considerable limitations of the proposed model, specifically with regards to it conceptual and empirical foundations.

From a conceptual point of view, the key 'psychological' mechanism through which the model explains increased obsessions seems to rely on a somewhat peculiar logic. As the reviewers now clarify, the model explains excessive 'entering into an obsessional state' as driven by the overgeneralization of the pleasantness of relief from obsession'. Such motivation for obsessions seems strange, and can be equated, in a different context, to a model that chooses to 'put its hand in the fire because it 'overgeneralizes the pleasantness of eventually taking it out of the fire'.

Furthermore, as the authors now clarify, this behavior seems to depend on several seemingly arbitrary settings of the simulations. Specifically, the model's choice to enter the obsessive state depends on the lack of enough alternative actions, and sufficient stochasticity (i.e., temperature). Along the same lines, I would assume that if the value of the other options would increase (i.e. if the model had additional, *rewarding enough* actions to choose from), this would also eliminate the obsessive behavior of the model. So, in other words, this model in the illustrative 'fire' context would 'randomly decides to put its hand in the fire because it doesn't have enough rewarding (interesting?) alternatives, and because the suffering entailed by this behavior is overweighed by the relief of eventually taking the hand out of the fire'.

Of course, the fact that this logic appears peculiar to me is not conclusive evidence against it. However, I believe the authors also do not provide sufficient empirical evidence for the model. Yes, the model can explain the results of the delayed feedback task, but so can the original 'eligibility trace' model. Yes, the model predicts reduced model-based behavior, but so does a 'neutral' SR-IR model (as correctly raised by Reviewer 3, and agreed to by the authors), and as it seems from Figure 9, an aversive SR + appetitive IR model. This latter model also appears to explain better the data from Voon (ref 7 in the manuscript), and after some speculations, the Gillan data (ref 9 in the manuscript). But, while this reversed model can, it seems, explain empirical data (as a side note, I think fitting the model to the Gillan data will be a much stronger proof here), whether it will predict obsessive behavior in the environment the authors use is questionable, and to the very least – should be examined.

Reviewer #3: I commend the authors for having done substantial work in their revision. In regard to my specific critiques, I think the simulation of Sakoi et al., is mostly compelling. I also think the authors have addressed my concern about why the SR weights diverge. However, my concerns over the two-step task simulations have increased.

The key claim of the paper is that OCD symptoms could be generated by a model combining SR and IR with asymmetric learning rates for positive and negative prediction errors, where SR learns more from appetitive prediction errors and IR learns more from aversive prediction errors. I commend the authors for looking at the Gilan et al., 2016 data to see to what extent this model is supported in two-step task data. I have some uncertainty about the test that was used to look for evidence that SR and IR have different learning rates for positive and negative prediction errors. I think a more straightforward approach would be to fit the model to the task data, to treat the learning rate for either system as free parameters, and then analyze how those learning rates change as a function of self-reported OCD symptoms. This could potentially also support the suggestion that healthy participants are described by SR alone (through model comparison). I found this surprising, since SR is typically thought to stand in for the model-based system, but not the model-free system, and healthy participants in the task are described by a mixture of model-based and model-free systems. I think the ability of the SR in simulations here to generate mixtures over MB and MF weightings might be due to that it learns the transitions - so this would be similar to the MB system learning the transitions from experience. This is a reasonable hypothesis for what generates behavior that might look like model-free learning (low w), and this this could potentially be supported by actual model-fitting.

More generally though, if the test used currently is correctly identifying learning rates for either system, it falsifies the model's predictions, instead showing that MB (or SR) behavior in OCD individuals is more influenced by negative than positive prediction errors relative to MF (or IR) behavior. The paper suggests an explanation for this. First, it argues that the Gilan et al. experiment, due to low pay, should be considered to be in the punishment rather than reward domain. Second, it argues that the punishment domain might encourage opposite learning rate asymmetries - where SR would learn more from negative errors and IR would learn more from positive (aversive SR + appetitive IR).

I did not find this argument to be convincing. In particular, the simulations of obsessive compulsive cycles (Fig. 2) take place in a punishment domain, yet despite this, the model used is appetitive SR + aversive IR. Additionally, the simulations demonstrate that aversive SR + appetitive IR in this domain (which they claim punishment domain might encourage) does not generate obsessive compulsive cycles. So, if I understood, the proposed model to explain the two-step task behavior is in conflict with the model used to explain obsessive compulsive cycles.

So, altogether, I’m somewhat uncertain about the extent to which the proposed model is supported as a model for OCD. The model of OCD decisions and variation from controls consists of two parts - 1) OCD participants use more IR component than healthy controls (both use an SR component), and 2) These components have different learning rates for positive and negative PEs, where SR is appetitive and IR is aversive. For the two-step task data, 1) is supported, but 2) is not. However, it’s worth noting that 1) is not really a new prediction for this task. Because it is known that SR and IR can stand in for MB and MF, the new account is not really different from the standard account of this data, which is that there is a shift from MB to MF control in OCD.

In contrast, to generate the obsessive compulsive cycles, it is really 2) that is needed, but not 1). That is, the key feature needed to explain variation between healthy controls and OCD patients in developing OCD cycles is imbalanced learning rates, not differences in amount of IR in addition to SR. So, in this regard, if the point of the paper is to offer a new model of generation of obsessive compulsive cycles, I’m not sure that the two-step task data is really offering support to the key feature of that model that is needed to support these cycles.

Overall, I do think the basic observation that some phenomena (the simulations of obsessive compulsive cycles and the data of Sakai et al.) which previously were argued to support mechanism of imbalance eligibility traces, could also be explained by a combination SR + IR learner with imbalanced learning rates is interesting still. But I’m not really satisfied with how the falsification of, what I view as the key part of this model, in the two-step task data, is explained.

**Have the authors made all data and (if applicable) computational code underlying the findings in their manuscript fully available?**

Reviewer #1: Yes

Reviewer #2: Yes

Reviewer #3: Yes

PLOS authors have the option to publish the peer review history of their article (what does this mean?). If published, this will include your full peer review and any attached files.

Reviewer #1: No

Reviewer #2: No

Reviewer #3: No
---

## [Editor Report · Decision Letter 2]

23 May 2023

Dear Dr Morita, 

We are pleased to inform you that your manuscript 'Opponent Learning with Different Representations in the Cortico-Basal Ganglia Pathways Can Develop Obsession-Compulsion Cycle' has been provisionally accepted for publication in PLOS Computational Biology.

Best regards,

Stefano Palminteri

Academic Editor

PLOS Computational Biology

Daniele Marinazzo

Section Editor

PLOS Computational Biology

---

## [Editor Report · Acceptance letter]

5 Jun 2023

PCOMPBIOL-D-23-00023R2 

Opponent Learning with Different Representations in the Cortico-Basal Ganglia Pathways Can Develop Obsession-Compulsion Cycle

Dear Dr Morita,

I am pleased to inform you that your manuscript has been formally accepted for publication in PLOS Computational Biology. Your manuscript is now with our production department and you will be notified of the publication date in due course.

With kind regards,

Timea Kemeri-Szekernyes
